# AnimeSR: Learning Real-World Super-Resolution Models for Animation Videos

**Yanze Wu**[*1]     **Xintao Wang**[*†1]     **Gen Li**[2]     **Ying Shan**[1]
[1]ARC Lab, Tencent PCG     [2]Platform Technologies, Tencent Online Video
{yanzewu, xintaowang, enochli, yingsshan}@tencent.com

## Abstract

This paper studies the problem of real-world video super-resolution (VSR) for animation videos, and reveals three key improvements for practical animation VSR. **First**, recent real-world super-resolution approaches typically rely on degradation simulation using basic operators without any learning capability, such as blur, noise, and compression. In this work, we propose to learn such basic operators from real low-quality animation videos, and incorporate the learned ones into the degradation generation pipeline. Such neural-network-based basic operators could help to better capture the distribution of real degradations. **Second**, a large-scale high-quality animation video dataset, *AVC*, is built to facilitate comprehensive training and evaluations for animation VSR. **Third**, we further investigate an efficient multi-scale network structure. It takes advantage of the efficiency of unidirectional recurrent networks and the effectiveness of sliding-window-based methods. Thanks to the above delicate designs, our method, *AnimeSR*, is capable of restoring real-world low-quality animation videos effectively and efficiently, achieving superior performance to previous state-of-the-art methods. Codes and models are available at https://github.com/TencentARC/AnimeSR.

## 1 Introduction

As a special category of video super-resolution (VSR) [5, 41, 47, 6], animation VSR aims at recovering high-resolution (HR) animation videos from their low-resolution (LR) counterparts. With the advent of the 4K/8K HD Era, people enjoy watching classic animation videos on HD devices. However, existing animation videos are mostly of low quality and have disturbing artifacts, which are usually caused by the production aging, lossy compression and transmission. Hence, it is highly desirable to develop practical animation VSR techniques to improve the resolution and quality of existing animation videos. Directly applying existing real-world SR methods (*e.g.*, Real-ESRGAN [48], RealBasicVSR [8]) produces unsatisfying and unnatural results with artifacts, such as "hollow line" artifacts, annoying noises, *etc*, as shown in Fig. 1 and Fig. 5.

In this paper, we thoroughly study and improve the key components in animation VSR.

• **Degradation synthesis process**. Real-world animation VSR is a blind SR task, whose LR videos have unknown and complex degradations, *e.g.*, blur, noise and compression artifacts. Recent methods sort to simulate degradations as close to the real-world ones as possible. The methods can be classified into two categories. One is to employ several classic basic operators together to form a complex degradation generation process [53, 48]. Those basic operators are usually "atomic" and have a clear physical meaning, including blur filters, noise generators, JPEG/FFMPEG compressor, *etc*. However, relying on simple basic operators without any *learning capability* largely limits its synthesis ability of real-world degradations.

The other category is to incorporate neural networks to synthesize LR samples with cycle consistency [57, 52, 13, 50] or self-supervision based on patch recurrence [58, 2]. Nevertheless, it is very

---

[*]Equal contributions. [†] Corresponding author

36th Conference on Neural Information Processing Systems (NeurIPS 2022).

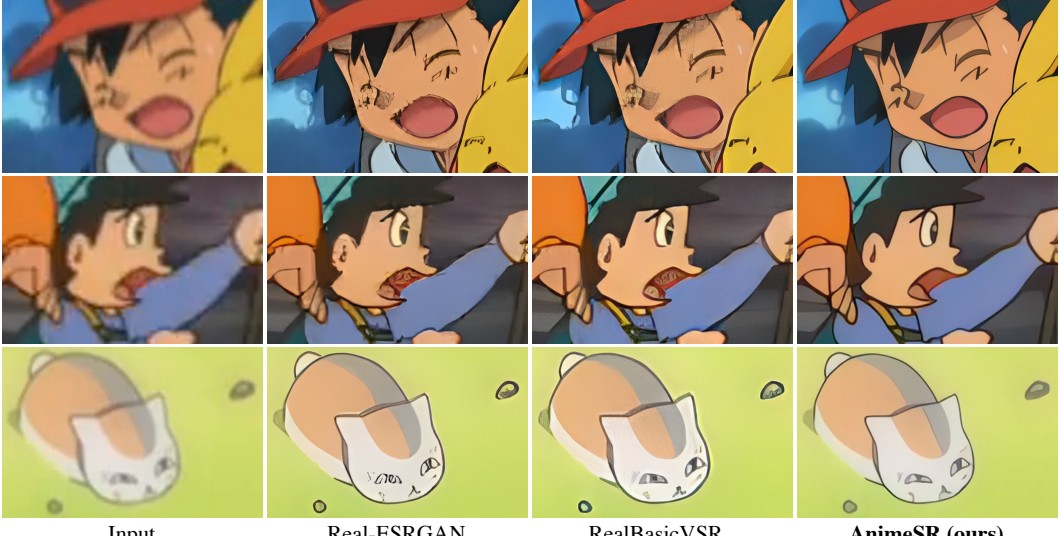

| Input | Real-ESRGAN | RealBasicVSR | **AnimeSR (ours)** |

Figure 1: **Comparisons of inputs, Real-ESRGAN [48], RealBasicVSR [8], and our AnimeSR results on real-world low-quality animation video frames**. The AnimeSR model is capable of restoring more natural details while suppressing disturbing noises and artifacts.

challenging to adopt *one large neural network* to learn the *whole* degradation process and the *entire* complicated degradation distribution. Thus, those methods only work for a very limited range of images and usually produce unpleasant artifacts.

In this work, we explore combining the above two categories and propose to **learn basic operators** for degradation synthesis. Specifically, we train *tiny neural networks* with 2 or 3 convolution layers to capture the main characteristics of real degradations. Those neural networks are then treated as basic operators, similar to blur filters or noise generators, to form the whole degradation synthesis process. The neural-network-based basic operators are learnable and are able to synthesize real degradations that those classic ones cannot model. A simple illustration is shown in Fig. 3.

Learning accurate and practical basic operators is still challenging, due to the lack of LR-HR paired data. Interestingly, we observe that it is possible to obtain pseudo HR samples for animation videos. For a real LR video, we can resize the input into different resolutions and obtain several HR outputs by existing models (*e.g.*, trained with classic operators). A satisfying output can be manually selected among those outputs. This is mainly due to the characteristics of animation videos, which usually consist of lines, sketches, and segments of smooth color pieces. More details are in Sec. 4.

• **High-quality animation video dataset**. The lack of a high-quality (HQ) animation video dataset impedes the development of animation VSR. Thus, we build a large-scale HQ Animation Video Clip dataset, namely **AVC**. Unlike previous animation datasets that either consist of only single images or low-quality frames [43], AVC contains 553 HQ clips with a total of 55,300 frames for training. Those clips are selected from various animation videos with different styles. Apart from the training set, we also build a real-world low-quality video test set from the Internet for practical evaluation. More descriptions and the statistics of AVC are provided in Sec. 3.

• **Efficient network structure for VSR**. For practical VSR, the unidirectional recurrent structure is usually adopted [41]. However, the unavailability of subsequent frames in unidirectional structure hinders the exploitation of temporal information. Similar to [14], based on the *efficient unidirectional structure*, we further utilize the *effectiveness of sliding-window structures* [47] in animation VSR. In order to fully exploit feature fusion at different scales, we also adopt the multi-scale design inside unidirectional recurrent blocks to improve the performance.

The contributions are summarized as follows. **(1)** We propose to learn basic operators for degradation synthesis, which better capture real-world degradations for animation videos. **(2)** A large-scale high-quality animation video dataset, AVC, is built for training and evaluations. **(3)** We investigate an efficient multi-scale network structure, which takes advantage of the efficiency of unidirectional recurrent networks and the effectiveness of sliding-window-based methods. **(4)** Equipped with the above improvements, our method, **AnimeSR**, is capable of restoring real-world low-quality animation videos effectively and efficiently, surpassing recent state-of-the-art methods.

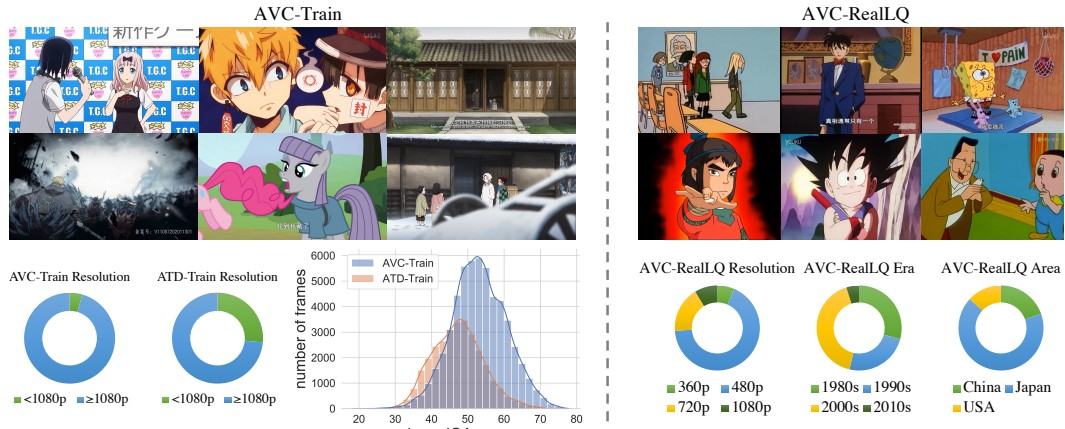

Figure 2: **Representative samples and statistics of AVC-Train and AVC-RealLQ sets**. AVC-Train contains clips with larger resolutions and higher quality than the ATD-12K [43] dataset. AVC-RealLQ consists of diverse low-quality clips with various resolutions from different years and areas.

## 2 Related Work

**Video Super-Resolution** (VSR) has attracted increasing attention recent years. Different from image super-resolution [11, 24, 26, 17, 27, 29, 55, 49, 54, 32, 35], VSR further explores the temporal fusion and propagation scheme. Existing VSR methods can be roughly divided into two categories: sliding window-based methods [47, 5, 45, 19, 28] and recurrent-based methods [6, 7, 14, 20, 21, 18]. Most previous methods focus on bicubic downsampling as the degradation. Recent works have started to address the real-world blind setting, such as DBVSR [39] and RealBasicVSR [8]. In this work, we aim to address the practical VSR for animation videos, which previous works have not investigated.

**The degradation model** used in existing SR methods can be divided into two categories: explicit degradation model [12, 31, 48, 53] and implicit degradation model [52, 13, 4, 34, 50, 56, 2, 10]. The explicit degradation model consists of basic degradation operators such as blur, noise and downsampling. It has developed from the early simple bicubic downsampling kernel to a classic sequence of blur, noise and downsampling, and finally to the current powerful high-order [48] and random-shuffled [53] models. The implicit degradation model attempts to learn the real-world degradation process with a neural network. Due to the lack of real-world LR-HR pairs, recent methods either use unpaired data with cycle consistency [57, 52, 13, 4, 34, 50, 56] or self-supervision based on patch recurrence property [58, 2, 10] to train the network. However, these methods only work for a limited range of images and usually produce unpleasant artifacts [30]. In this paper, we explore combining both the explicit and implicit degradation models.

**Animation** in this paper refers specifically to animated cartoons featured with exaggerated visual style. Various animation processing techniques using deep learning have been proposed in recent years. For example, animation style transfer [9], animation video interpolation [43], animation video generation for full-body characters [16]. However, existing animation datasets are usually of low quality and contain only single images or triplet frames. These shortcomings make them unsuitable for animation VSR, where HQ and long frame sequences are required. Thus, we build a large-scale HQ animation video clip dataset to facilitate the training and evaluation of VSR models.

## 3 Animation Video Clip (AVC) Dataset

To facilitate the training and evaluation of VSR methods on animation videos, we build a large-scale high-quality **A**nimation **V**ideo **C**lip dataset, namely, AVC dataset. AVC is split into three sets. The training set, *AVC-Train*, contains 553 high-quality clips with a total of 55,300 frames. The testing set, *AVC-Test*, contains 30 clips with a total of 3,000 frames. The AVC-Test has a similar quality to AVT-Train, but are excluded from the training set. In order to evaluate the methods in practical scenarios, we also build a real-world testing set, *AVC-RealLQ*, which consists of 46 low-quality clips from the Internet. Some examples of the AVC dataset are shown in Fig 2.

**Dataset Collection of AVC-Train and AVC-Test.** To ensure the **high quality** of the dataset, we first download a large number of animation videos from the Internet and then manually select high-quality ones based on bit rate, frame resolution, and subjective quality. We use FFmpeg to extract frames

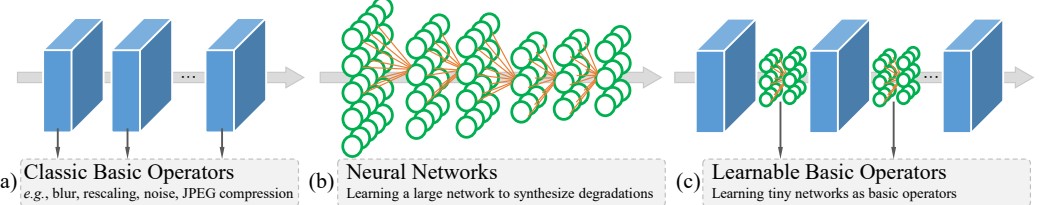

Figure 3: **Different ways to synthesize degradations**. **(a)** Typical degradation synthesis uses pre-defined basic operators, such as blur, rescaling, noise and JPEG compression. **(b)** Neural networks are also employed to learn degradations. **(c)** We propose to learn tiny networks as basic operators.

with q:v setting to 1. We also adopt the hyperIQA [44], an image assessment quality algorithm, to calculate the quality score for each frame. Frames with low hyperIQA scores are discarded. To select the **dynamic and meaningful scenes**, PySceneDetect[1] is employed to detect and split scenes. Optical flow [40] is calculated to filter out static scenes. We keep a few clips with scene transitions, so that the trained models are more robust to real-world animation videos with scene transitions. To avoid redundancy and improve **diversity**, we only keep one clip for each video. The selection criterion is a weighted evaluation of hyperIQA scores and optical flow values. We always select high-quality frames with apparent motions. Following the practice in [38], the total frame number for each clip is set to 100. After this selection, we obtain 583 high-quality clips, and partition 553 clips for training and 30 clips for testing.

**Dataset Statistics.** The resolution and hyperIQA distributions of the AVC dataset are shown in Fig. 2. We also compare our AVC dataset with the recently proposed ATD-12K [43] to demonstrate the superiority of the AVC dataset. Our AVC dataset contains clips with larger resolutions than ATD-12K. HyperIQA scores reflect the perceptual quality, whose values are between 1 and 100. The higher the scores are, the better the quality is. We can observe that the AVC dataset also has higher quality than ATD-12K. Besides, recent VSR methods (*e.g.*, BasicVSR [6]) require long sequences (*e.g.*, 15 frames) during training to better leverage long-term propagation. However, ATD-12K with only triplet frames is not suitable for the VSR task.

**AVC-RealLQ.** To fully evaluate the model's ability in practical scenes and assess the generalizability of VSR methods, we also construct a real-world low-quality video test set from the Internet, namely, AVC-RealLQ. Those videos are downloaded from various video sites such as Bilibili, Youtube, and Tencent Video. The different resolutions, contents, and genre are considered for the diversity. We collect 46 real-world low-quality clips, with each clip containing 100 frames. The representative samples and statistics are shown in Fig. 2.

## 4 Methodology

### 4.1 Learning Basic Operators for Degradation Synthesis

Real-world animation video super-resolution (VSR) aims to restore high-resolution videos from low-resolution ones with unknown and complex degradations, *e.g.*, blur, noise and downsampling. Due to the lack of LR-HR training pairs, recent works sort to design degradation models as close to the real-world degradations as possible. Then, LR samples are synthesized from available HR samples using the degradation model. The degradation model can be described as an $n$-step degradation process and formulated as:

$$\boldsymbol{x} = \mathcal{D}^n(\boldsymbol{y}) = (\mathcal{D}_n \circ \cdots \circ \mathcal{D}_2 \circ \mathcal{D}_1)(\boldsymbol{y}), \qquad (1)$$

where $\boldsymbol{y}$ is an HR input, $\boldsymbol{x}$ is the synthetic LR output, and $\mathcal{D}_i$ is the $i$-th basic degradation operator.

The classical degradation model is a four-step process, in which basic operators include blur filters, noise generators, rescaling, JPEG/FFMPEG compressor, *etc*. High-order degradation process [48] and shuffling strategy [53] are then proposed to expand the degradation space, so that it can cover more practical degradations. However, the degradation process still involves those classic basic operators only. We denote this category as BasicOp-Only, as illustrated in Fig. 3 (a). The basic operators are "atomic" and do not have any learning ability, which essentially limits their synthesis ability of real-world degradations. Another category adopts large neural networks (NN) and adversarial learning to synthesize LR samples. We denote this category as NN-Only, as shown in Fig. 3 (b). However, it is challenging to employ one large neural network to learn the *whole* degradation process and the *entire* complicated degradation distribution. As a result, those methods only work for a limited range of images and usually produce unpleasant artifacts.

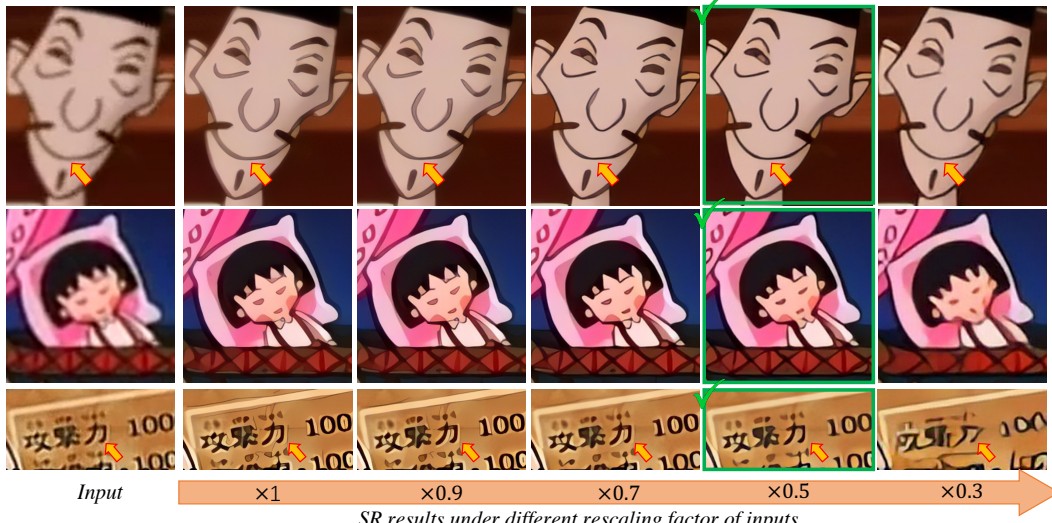

| Input | ×1 | ×0.9 | ×0.7 | ×0.5 | ×0.3 |

*SR results under different rescaling factor of inputs*

Figure 5: **Illustration of the "input-rescaling strategy"**. We resize the inputs with different rescaling factors, *i.e.*, from ×1 to ×0.3, and generate the SR results using a VSR model trained with `BasicOP-Only` degradations. We can observe "hollow line" artifacts, annoying noises and unwanted textures for ×1 rescaling. The artifacts are gradually reduced as the input resolution decreases. Rescaling the inputs by ×0.5 produce a satisfying output. **Zoom in for best view**

As shown in Fig. 3 (c), we propose to **learn basic operators** for degradation synthesis. Unlike `NN-Only` methods that use one large network, we train *tiny neural networks* with 2 or 3 convolution layers to capture the main characteristics of real degradations. Those neural networks are then incorporated with the classic basic operators to form the degradation model. The neural operators are learnable and are able to synthesize real degradations that cannot be modeled by those classic ones. As illustrated in Fig. 4, with the learnable basic operator, the degradation space is largely expanded and can cover more real degradations.

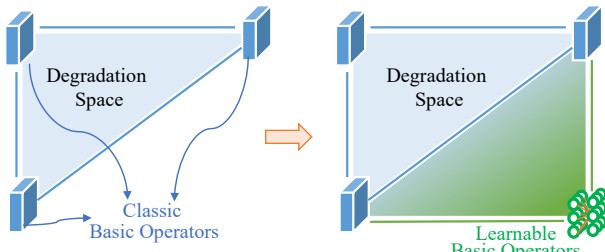

Figure 4: **The degradation space "spanned" by basic operators**. The learnable basic operators expand the degradation space with more real degradations.

**How to obtain learnable basic operators?** Different to `NN-Only` methods that train with adversarial learning and unpaired data, we train learnable basic operators with LR-HR pairs in a supervised manner to learn the mappings from HR to LR. Such a strategy is more controllable and introduce fewer artifacts. Yet, it is challenging to get LR-HR pairs of real-world low-quality (LQ) videos for training. Our analysis below shows that it is possible to obtain pseudo HR samples for real animation videos.

For real LQ animation videos, we can easily obtain the preliminary SR results using a VSR model trained with commonly-used `BasicOP-Only` degradations, as shown in the second columns of Fig. 5. As expected, the outputs are unsatisfying. We can observe "hollow line" artifacts, annoying noises and unwanted textures in the second columns of Fig. 5. We then resize the inputs with different rescaling factors, *e.g.*, from ×1 to ×0.3. The rescaled inputs are sent to the same VSR model, the corresponding SR results are presented in Fig. 5 (the third column to the last column). It can be observed that the artifacts are gradually reduced as the input resolution decreases. But too large downscaling factor on inputs leads to detail/information loss. Among them, rescaling the inputs by ×0.5 on those video samples produce a good trade-off between artifacts elimination and detail loss. Thus, a **satisfying output can be manually selected as the pseudo HR**. We call this simple yet effective strategy as **"input-rescaling strategy"**. Note that the best rescaling factor can vary for different real-world animation videos. More details about the "input-rescaling strategy" can be found in the supplementary material.

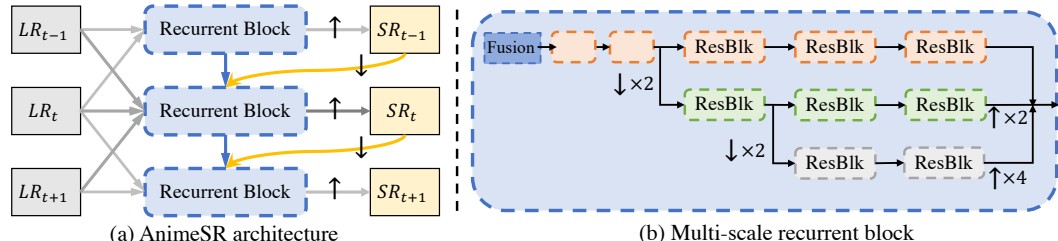

|  |  |
|---|---|
| (a) AnimeSR architecture | (b) Multi-scale recurrent block |

Figure 6: **The network structure of AnimeSR**. (a) AnimeSR combines the efficiency of unidirectional recurrent networks and the effectiveness of sliding-widow-based methods. (b) The recurrent block employs a multi-scale design to fully exploit the capability for animation videos.

**Why the "input-rescaling strategy" works for animation video?** The "input-rescaling strategy" is able to *eliminate artifacts* while *not affecting the details* under a proper rescaling factor. We hypothesize that: **1).** the degradation of real LQ videos is probably outside the synthesized degradation space "spanned" by classic basic operators. Rescaling the inputs changes the degradation and may narrow its gap towards synthesized degradation space. Besides, downscaling also makes the LR samples look "cleaner", which is also pointed in [13]. **2).** Animation videos usually consist of lines, sketches, and segments of smooth color pieces, which have very different characteristics from natural videos. The particular characteristics of animation videos lead to a phenomenon that rescaling within a limited range will not have much influence on details. This strategy cannot be directly applied in natural videos, as it will largely affect the textures.

**Details of training learnable basic operators.** We pick out several representative real-world LQ animation videos to train learnable basic operators. We select LQ videos, on which the VSR model performs poorly on the original scale, but produce satisfactory results on a suitable rescaling factor. Then, we determine the best rescaling factor of inputs for each video. After that, about 2000 frames are collected for one LQ video. Similar to the collection pipeline of the AVC dataset, the motions and scene diversity are considered during the collection. We then resize those frames with the best rescaling factor, feed them into the VSR networks, and finally obtain pseudo HR samples.

The LR-pseudo HR pairs are then used to train the learnable basic operators. The tiny neural operator is lightweight and is composed of three $3 \times 3$ convolution layers with the hidden channel dimension of $64$. LeakyReLU activations are employed between convolution layers. The training objective of neural operators is the same as that for the main network structure, as described in Sec. 5.1.

**Discussions. 1**). We observe that learned operators have strong generalization. An operator trained on one LQ video can also be effective for other LQ videos. This property largely improves its practicability. We hypothesize that the captured real degradations probably have commonality, and cannot be well modeled by classic basic operators. **2**). Unlike the classical basic operator with a clear physical meaning for each one, what the neural operator learns does not have a clear definition. It may capture a mixture of blur and noise, reflecting the main characteristics of the real degradation in LQ videos. The visualization of leaned basic operators is demonstrated in Sec. 5.3. **3**). In this paper, we train **three** learnable basic operators from three different LQ videos respectively and put them into a pool. During each training iteration, we randomly select one from the pool and directly incorporate it into the degradation generation pipeline. This is a simple incorporation design of learnable basic operators. More schemes can be explored in future works.

### 4.2 Network Structure

The network structure in practical animation VSR requires a good balance between performance and efficiency. Recent practical models such as BSRGAN [53], Real-ESRGAN [48] and RealBasicVSR [8] usually adopt a very large network, whose processing is time-consuming and resource-intensive. This shortcoming becomes more severe when super-resolving existing videos to 4K/8K resolutions. For practical VSR, the *unidirectional recurrent structure* is usually adopted [41]. Yet, the unavailability of subsequent frames hinders the exploitation of temporal information. Therefore, based on the efficient unidirectional structure, we further employ the simple sliding-window structure [47].

The overview is depicted in Fig. 6 (a). The upsampling operation is performed at the last layer with one pixel-shuffle layer [42] to obtain SR outputs, so that most of the calculation is conducted in the LR feature space. At the time $t$, the recurrent block receives the information from the previous hidden state $S_{t-1}$ of the recurrent block, and the SR output of time $t - 1$. The output $SR_{t-1}$ is

of high resolution and is downsampled to match the spatial size of the recurrent block using a pixel-unshuffle layer. The recurrent block also receives a sequence of $\{LR_{t-1}, LR_t, LR_{t+1}\}$ frames, as sliding-window-based methods do.

As analyzed in Sec. 4.1, rescaling inputs of animation videos achieve a better balance of detail enhancement and artifacts elimination. Thus, we also adopt the multi-scale design inside the unidirectional recurrent block. Besides, the multi-scale structure can also help to fully exploit feature fusion at different scales. As shown in Fig. 6 (b), we adopt 10 residual blocks in the recurrent block. We employ three scales, *e.g.*, $\times 1$, $\times 0.5$ and $\times 0.25$. We allocate 5, 3 and 2 blocks for those three scales, respectively. We do not employ optical flow in our AnimeSR, as we empirically find that optical flow does not bring apparent visual improvements. Besides, calculating optical flow also slow down the training and inference.

## 5 Experiments

### 5.1 Training Details

The training process consists of two stages. In the first stage, we train models with the L1 loss for $300K$ iterations. In the second stage, we finetune the models for another $300K$ iterations with L1 loss, perceptual loss [23], and GAN loss [27], whose loss weights are all set to 1. The discriminator is a three-scale PatchGAN [22, 46] with spectral normalization [37]. We use the Adam optimizer [25] with a learning rate of $2 \times 10^{-4}$ for the first stage and a learning rate of $1 \times 10^{-4}$ for the second stage. We set the batch size per GPU, frame sequence length, and patch size of the HR frames to 4, 15, and 256, respectively. All the training is performed with PyTorch on four NVIDIA A100 GPUs in an internal cluster.

To obtain the learnable basic operators, we first train a VSR model with a degradation model that only contains the classic basic operators (Fig. 3 (a)). Then we utilize this model and "input-rescaling strategy" to generate the pseudo HR samples. The training strategy for the learnable basic operators is almost the same as mentioned above, except that the batch size and iteration are set to 16 and $100K$. The final VSR model is trained with a degradation model containing both classic basic operators and learnable operators. Specifically, the degradation model can be formulated as:

$$\boldsymbol{x} = \mathcal{D}^n(\boldsymbol{y}) = (\texttt{FFmpeg} \circ \texttt{LBO} \circ \texttt{Down} \circ \texttt{LBO} \circ \texttt{Down} \circ \texttt{Noise} \circ \texttt{Blur})(\boldsymbol{y}), \qquad (2)$$

where $\boldsymbol{y}$ is an HR frame sequence, $\boldsymbol{x}$ is the synthetic LR frame sequence. LBO denotes the Learnable Basic Operator. Blur represents blur filters, and we employ isotropic and anisotropic Gaussian kernels with a probability of $\{0.7, 0.3\}$. Noise denotes the Gaussian color noise and Gaussian gray noise with a probability of $\{0.5, 0.5\}$. Down denotes the downscaling operation with a scale factor of 2. Finally, we use FFmpeg to perform video compression. More details about the hyper-parameters are provided in the supplementary materials.

### 5.2 Comparisons with Previous Methods

We compare our method with recent state-of-the-art methods for real-world SR, including BSR-GAN [53], Real-ESRGAN [48] and RealBasicVSR [8]. For a fair comparison, we *finetune* their officially released models on our proposed AVC-Train dataset. We also provide the comparisons before and after finetuning in the supplementary material.

**Quantitative comparisons.** All methods are evaluated on the real-world AVC-RealLQ dataset. As ground-truths are not available for real-world LQ animation videos, we use no-reference image quality assessment (NR-IQA) metrics NIQE [36] and MANIQA [51] to evaluate the methods quantitatively. NIQE is widely used for the evaluation of real-world SR models [53, 48, 8], it is based on natural scene statistics and hand-crafted features. MANIQA is the winner solution of the NTIRE 2022 NR-IQA challenge [15], it is a learning-based approach and employs a powerful ViT to extract features. The comparison results are shown in Tab. 1. We train two AnimeSR models with one and three learnable basic operators. Our method achieves better NIQE and MANIQA scores with *significantly smaller model size and faster processing speed* (*i.e.*, $13.7\times$ faster than BSRGAN and $5.9\times$ faster than RealBasicVSR). The high efficiency makes our AnimeSR a more practical animation VSR model. As many previous works [3, 33, 53] have pointed out, NR-IQA metrics are not always consistent with visual quality, especially on the finer scale. Empirically, we find that the recent MANIQA [51] is more consistent with the perceptual visual quality than NIQE.

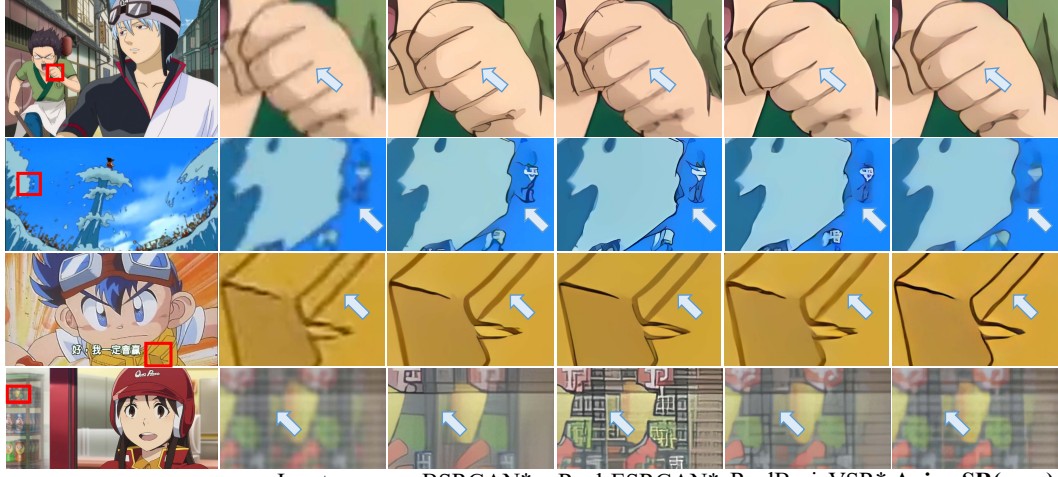

| | Input | BSRGAN* | Real-ESRGAN* | RealBasicVSR* | **AnimeSR(ours)** |

Figure 7: **Qualitative comparisons**. '*' denotes fine-tuning on our AVC dataset. Our AnimeSR could recover cleaner lines and produce more visual-pleasing outputs, while having fewer artifacts. AnimeSR can also restore natural textures in blurred background, rather than impair those textures (4th row). **Zoom in for best view**

Table 1: Quantitative comparisons on AVC-RealLQ. *LBO* denotes the Learnable Basic Operator. '*' denotes fine-tuning on our AVC dataset. The runtime is computed on a Nvidia A100 with the LR size of $480 \times 640$. The I/O time is not included.

| | BSRGAN* [53] ICCV 21 | Real-ESRGAN* [48] ICCVW 21 | RealBasicVSR* [8] CVPR 22 | **AnimeSR** 1 *LBO* | **AnimeSR** 3 *LBO* |
|---|---|---|---|---|---|
| Params (M)↓ | 16.7 | 16.7 | 6.3 | **1.5** | **1.5** |
| Runtime (ms)↓ | 439 | 439 | 190 | **32** | **32** |
| NIQE↓ | 8.6369 | 8.3220 | 8.6393 | **8.1444** | 8.7088 |
| MANIQA↑ | 0.3753 | 0.3782 | 0.3602 | 0.3763 | **0.3832** |

**Qualitative comparisons.** We also show qualitative comparisons on real-world AVC-RealLQ video clips in Fig. 7. It is observed that our AnimeSR can recover cleaner and sharper lines with fewer artifacts (the 1st row and 3rd row), and produce more natural outputs in the fountain and water in the 2nd row, while previous methods generate wired lines and patterns. For the fourth row in Fig. 7, the input frame has a blurred background with a shallow depth of field. Previous methods either over-sharpen the background or impair the textures. While the AnimeSR is able to retain the overall naturalness and coherency. More visual comparisons are in the supplementary materials.

## 5.3 Ablation Studies and Discussions

Unless specified otherwise, we adopt one learnable basic operator (LBO) in ablations. Empirically, we find that the recent MANIQA [51] is more consistent with the perceptual visual quality than NIQE. Thus, we report the MANIQA score for ablation studies.

**Influence of different animation datasets.** We compare models trained with our AVC dataset and ATD-12K dataset. The model trained with AVC dataset could produce outputs with higher quality (Tab. 2) and cleaner lines, as shown in Fig. 8 (a).

**Effectiveness of the learnable basic operators.** We compare the following variants. `BasicOP-Only` denotes the VSR model trained without LBO. For `NN-Only`, we employ the degradation model proposed in DSGAN-SR [13]. DSGAN-SR employs a large network to learn the real-world degradations in an unpaired manner. The network has $594K$ parameters and is significantly larger than our tiny learnable basic operator. We re-train DSGAN-SR for a fair comparison. We also compare our AnimeSR with one LBO and three LBOs. The qualitative comparison is shown in Fig. 8 (b). We can observe that `BasicOP-Only` produces "hollow line" artifacts while `NN-Only` introduces extra artifacts around lines. Our method can clearly sharpen lines without annoying noises. Quantitative results in Tab. 2 also show the superiority of our degradation model with LBOs.

**Benefits of multi-scale architecture.** The effectiveness of the "input-rescaling strategy" motivates us to employ a multi-scale architecture. We compare our network structure with the single-scale one to validate its effectiveness. Both architectures have the same number of parameters. From Fig. 8 (c), we can find that the multi-scale architecture could produce cleaner lines and fewer artifacts than the single-scale architecture.

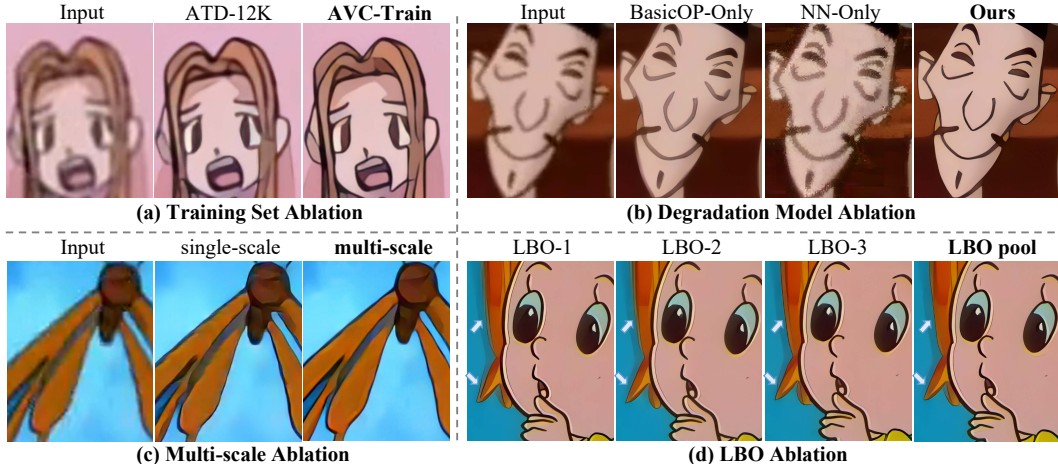

| Input | ATD-12K | **AVC-Train** | Input | BasicOP-Only | NN-Only | **Ours** |

**(a) Training Set Ablation**       **(b) Degradation Model Ablation**

| Input | single-scale | **multi-scale** | LBO-1 | LBO-2 | LBO-3 | **LBO pool** |

**(c) Multi-scale Ablation**       **(d) LBO Ablation**

Figure 8: **Visual comparisons of different ablation studies**. **Zoom in for best view**

Table 2: Ablation studies on the dataset, degradation model, multi-scale structure and the learnable basic operators (LBO). We report the MANIQA scores for comparison.

|  | **Training Set Ablation** |  | **Degradation Model Ablation** |  |  |  |
|---|---|---|---|---|---|---|
|  | ATD-12K | AVC-Train | BasicOP-Only | NN-Only | ours w/ 1 LBO | ours w/ 3 LBO |
| MANIQA↑ | 0.3309 | **0.3763** | 0.3554 | 0.3034 | 0.3763 | **0.3832** |
|  | **Multi-scale Ablation** |  | **LBO Ablation** |  |  |  |
|  | single-scale | multi-scale | LBO-1 | LBO-2 | LBO-3 | LBO pool |
| MANIQA↑ | 0.3283 | **0.3763** | 0.3763 | 0.3803 | 0.3796 | **0.3832** |

**Benefits of LBO pool.** We learn three learnable basic operators (we use LBO-1, LBO-2, and LBO-3 to denote the three different LBOs) from three representative LQ videos and then put them into a pool for random selection during training. We compare such a strategy with a single LBO. It is observed from Fig. 8 (d) that using a single LBO probably leads to ghosting artifacts and unclear lines in the curtains, while LBO pool can produce clean lines and sketches. Employing the LBO pool also achieve higher MANIQA scores from Tab. 2.

**Visualizations of what the learnable basic operators learn.** We visualize the LQ patches synthesizing from the same HQ image using different basic operators. As shown in Fig. 9, the same HQ input patch goes through five different basic degradation operators, generating five LQ patches. These degradation operators consist of two classic basic operators (*i.e.*, Gaussian blur and Gaussian noise) and three learnable basic operators (*i.e.*, LBO-1, LBO-2, and LBO-3). It is observed that the degradations learned by LBO are pretty different from classic basic operators, *e.g.*, Gaussian blur and Gaussian

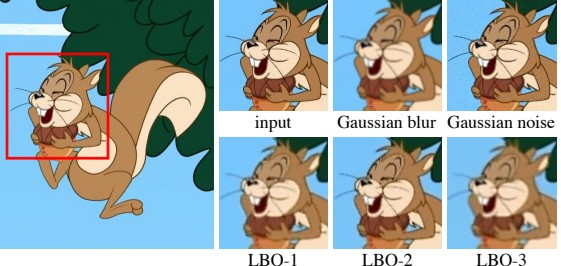

Figure 9: **LQ patches generated using basic operators.** Learnable basic operators capture diverse degradations that are different from the predefined ones (*e.g.*, blur and noise). **Zoom in for best view**

noise. It seems that LBO captures a mixture of blur and noise (here blur and noise stand for a vague human perception). We can also observe color jitter around lines in the LBO-1 neural operator, which is common in highly-compressed LQ videos. Besides, the three LBOs learn different degradations. The diversity of LBO makes it meaningful to combine multiple LBO during training.

**Limitations and Discussions.** Our work has several limitations. 1) The learning of basic operators in a supervision manner heavily relies on the "input-rescaling strategy". It fully exploits the characteristics of animation videos but cannot be directly applied to videos of natural scenes. 2) We are interested in building a larger pool of learnable basic operators from more real LQ videos. The complementarity and substitutability of these operators remain to be explored.

# 6 Conclusion

This paper gives a comprehensive study of the practical animation video super-resolution task. To facilitate the training and evaluation of animation VSR, we build a large-scale HQ animation video dataset named AVC. We also propose to learn basic operators in the degradation generation pipeline to better capture the distribution of real-world degradations. Learning of those neural operators is made possible by the effective "input-rescaling strategy" for real low-quality animation videos. We further investigate an efficient multi-scale network structure. Extensive experiments demonstrate the capability of our proposed method, AnimeSR, in restoring real-world low-quality animation videos, outperforming previous state-of-the-art methods.

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
