# Supplementary Materials for
# AnimeSR: Learning Real-World Super-Resolution Models for Animation Videos

**Yanze Wu**[*][1]     **Xintao Wang**[*][†][1]     **Gen Li**[2]     **Ying Shan**[1]

[1]ARC Lab, Tencent PCG     [2]Platform Technologies, Tencent Online Video

{yanzewu, xintaowang, enochli, yingsshan}@tencent.com

## Abstract

In this supplementary material, we provide:

- More details about the network structures in Sec. 1.
- More details about the "input-rescaling strategy" in Sec. 2.
- Hyperparameters of the degradation model in Sec. 3.
- Comparisons before and after finetuning on our AVC-Train dataset for state-of-the-art methods in Sec. 4.
- More visual comparisons on real-world low-quality animation videos with state-of-the-art methods in Sec. 5.

More details and demos can be found in https://github.com/TencentARC/AnimeSR.

## 1  Network Structure Details

**More details about the multi-scale recurrent block (MSRB).** The residual block in MSRB consists of two $3 \times 3$ convolution layers and one ReLU activation layer. The hidden channel dimension is set to $64$. For the downsampling operation between different scales of the MSRB, we employ a $3 \times 3$ convolution layer with the stride of 2. For the upsampling operation, we adopt a bilinear upsampling layer. After the features from different scales are upsampled to the same scale, we concatenate them and use two $3 \times 3$ convolution layers to acquire the SR frame and the recurrent state.

**More details about the learnable basic operator (LBO).** Given a sequence of LR frames $I_{LR} \in \mathbb{R}^{n \times h \times w \times c}$ from real-world animation video, where $n$ is the selected frames to train the LBO. $h$, $w$, and $c$ are the height, width, and channel number of the frame, respectively. Firstly, we select the best rescaling factor (usually around $0.5$) for a specific animation video. Secondly, we obtain the pseudo HR frames and resize them to $I_{pseudo\_HR} \in \mathbb{R}^{n \times 2h \times 2w \times c}$. To learn the mapping from pseudo HR to LR, LBO also contains a pixel-unshuffle layer with a scale factor of 2 to perform the downsampling at the beginning.

## 2  More Details about the Input-Rescaling Strategy

The input-rescaling strategy is **only** used in learning the learnable neural operators. The best rescaling factor is selected with a combination of algorithms and manual selection/verification. However, the algorithms' performance is not always satisfactory, especially for distinguishing the fine-grained artifacts. Thus, **manual** selection/verification is necessary.

The details of the rescaling factor selection for a specific real-world low-quality video are as follows.

---

[*]Equal contributions. [†] Corresponding author

36th Conference on Neural Information Processing Systems (NeurIPS 2022).

1. Select patches from the LQ video with textures and edges, because unpleasant artifacts are most likely to appear in those areas. The selection can be done with edge detection.

2. Evenly sample rescaling factors from $(0, 1]$ with an interval of $0.1$. For each rescaling factor, LR patches are rescaled and are then sent to the BasicOP-Only VSR model to get the SR results.

3. Sort the rescaling factor based on the number of artifacts. Empirically, we compare simple image statistics (*e.g.*, image gradients) for the assistant, followed by a manual selection.

4. Based on the sorting from Step 3, we manually select the best rescaling factor and pseudo HR according to human perception.

The selection is **only** performed for a few videos. Models trained with a few learnable neural operators can generalize well to a lot of real-world videos. In this work, the learnable neural operators from LQ videos can well generalize to a large number of real-world LQ videos.

# 3 Hyperparameters of the Degradation Model

As described in the main paper, the degradation model can be formulated as:

$$\boldsymbol{x} = \mathcal{D}^n(\boldsymbol{y}) = (\texttt{FFmpeg} \circ \texttt{LBO} \circ \texttt{Down} \circ \texttt{LBO} \circ \texttt{Down} \circ \texttt{Noise} \circ \texttt{Blur})(\boldsymbol{y}), \quad (1)$$

- For `Blur`, we employ isotropic and anisotropic Gaussian kernels, with a probability of $\{0.7, 0.3\}$. The standard deviation range of isotropic and anisotropic Gaussian kernels are set to $[0.2, 4.0]$ and $[0.8, 3.0]$, respectively.

- For `Noise`, we adopt Gaussian color noise and Gaussian gray noise with a probability of $\{0.5, 0.5\}$. The sigma range of noise is set to $[0, 10]$.

- `Down` denotes the downscaling operation with the scale factor of 2. We randomly choose one of the interpolation algorithms: area, bilinear, and bicubic.

- `LBO` denotes the learnable basic operator. For the two LBO in Eq. (1), the probability of each LBO being employed in the degradation process is $0.5$. Since the LBO already contains a pixel-unshuffle layer with a scale factor of 2, the previous adjacent `Down` operation will be skipped if the LBO is employed. The weights of two LBO in Eq. (1) are randomly selected from the LBO pool (3 in this paper). They can be with different weights, however, for ease of implementation, we let two LBO in Eq. (1) share the same weight.

- `FFmpeg` denotes the FFmpeg compression. The constant rate factor range is randomly chosen from $[18, 35]$. The H.264 profile is randomly select from [baseline, main, high] with a probability of $[0.1, 0.2, 0.7]$.

# 4 The Benefits of Finetuning on our AVC-Train Dataset

We provide the qualitative comparisons before and after finetuning on our AVC-Train dataset for the state-of-the-art (SOTA) methods: BSRGAN [3], Real-ESRGAN [2], and RealBasicVSR [1]. As shown in Fig. 1 - 3, after finetuning on the AVC-Train dataset, the annoying noise and artifacts are largely reduced. We can also observe that the lines after finetuning are cleaner.

# 5 More Qualitative Comparisons with SOTA methods

In this section, we provide more qualitative comparisons with the SOTA methods. Consistent with the settings in the main paper, the SOTA methods are finetuned on our AVC-Train dataset for fair comparisons. As shown in Fig. 4 - 14, our method can:

- recover cleaner and sharper lines with fewer artifacts (see Fig. 4 - 8).

- restore more texture details (see Fig. 9 - 11).

- produce more natural and coherent results for the background and blurred areas with a shallow depth of field (see Fig. 12 - 14 and Fig. 8).

**BSRGAN**

| Input | w/o finetune | w/ finetune |

Figure 1: The benefits of finetuning BSRGAN on our AVC-Train dataset. After fine-tuning on the AVC-Train dataset, the model can produce clearer lines with fewer artifacts, as denoted by the arrows in the figures. **Zoom in for best view**

**Real-ESRGAN**

| Input | w/o finetune | w/ finetune |
|---|---|---|

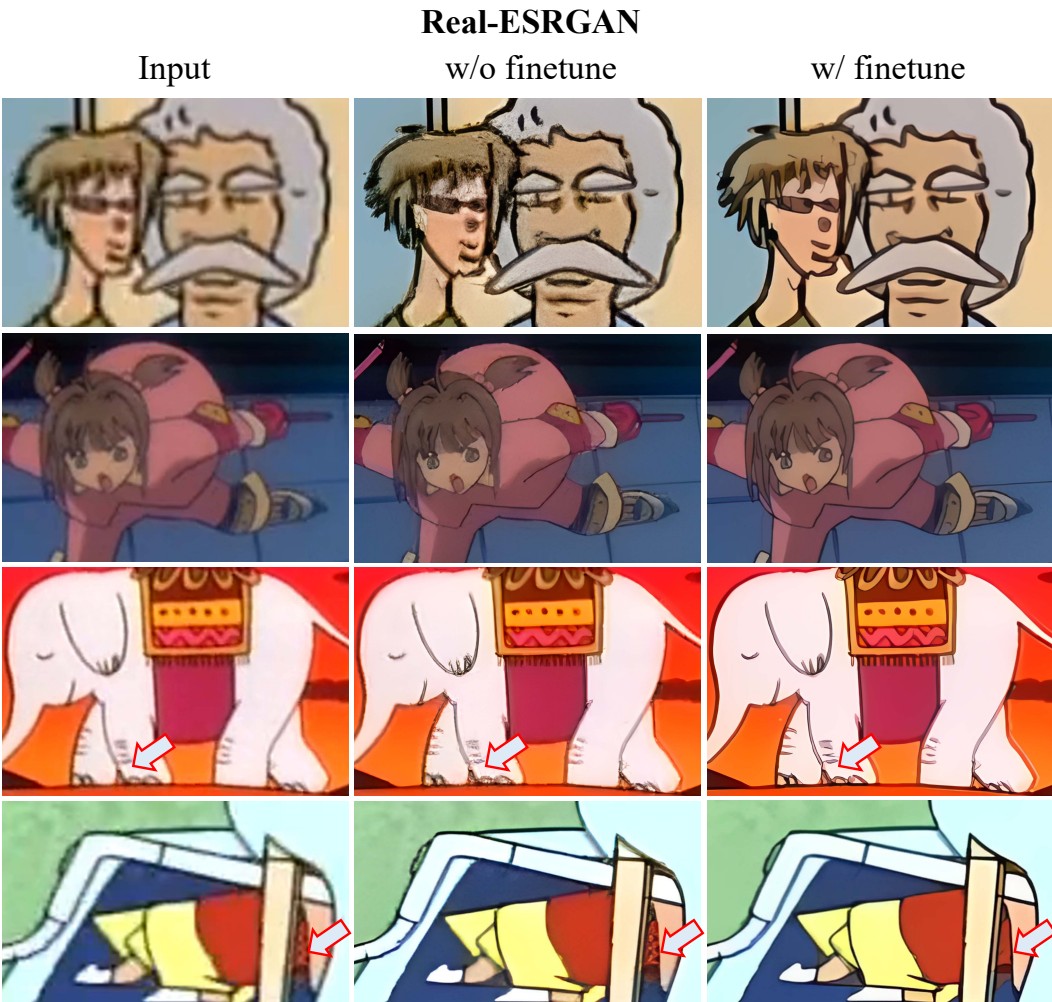

Figure 2: The benefits of finetuning Real-ESRGAN on our AVC-Train dataset. After fine-tuning on the AVC-Train dataset, the model can produce fewer artifacts (the characters in the first row and the elephant feet in the third row), clearer and sharper lines (the second row). It can also remove unpleasant noises (the fourth row). **Zoom in for best view**

**RealBasicVSR**

| Input | w/o finetune | w/ finetune |
|---|---|---|

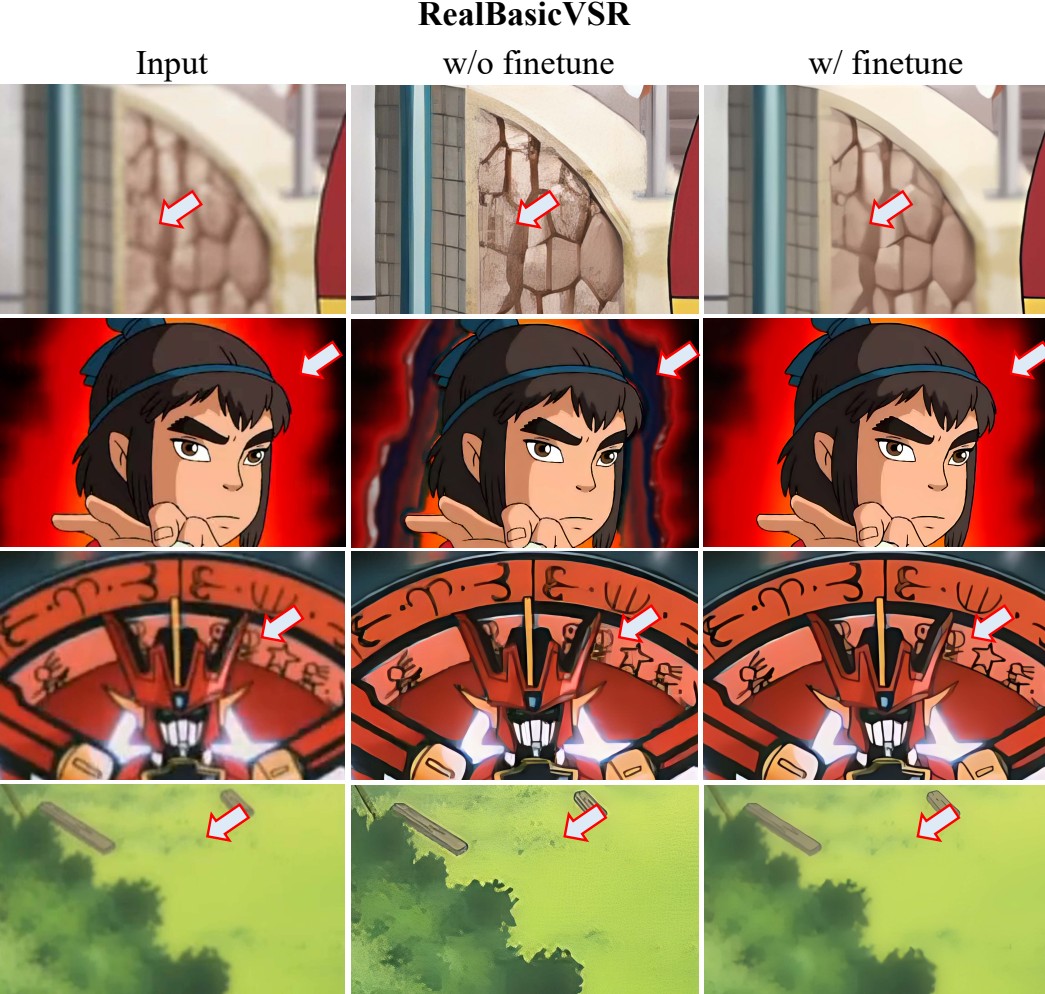

Figure 3: The benefits of finetuning Real-BasicVSR on our AVC-Train dataset. After fine-tuning on the AVC-Train dataset, the model can produce cleaner output with fewer artifacts (the bricks in the first row and the background in the second row), clearer and sharper lines (the third row). In the fourth row, the model without fine-tuning adds noises and artifacts on grass. **Zoom in for best view**

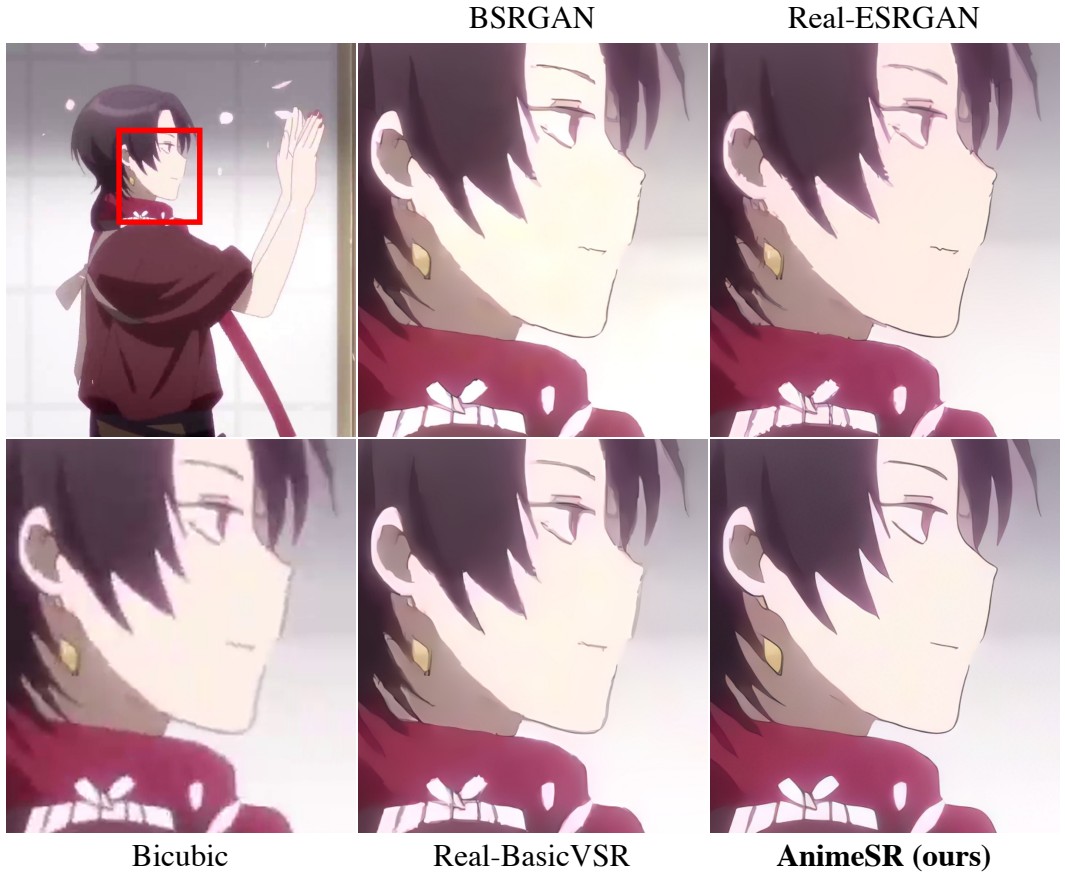

Figure 4: Qualitative comparison. Our method can recover cleaner lines with fewer artifacts around faces and hair. **Zoom in for best view**

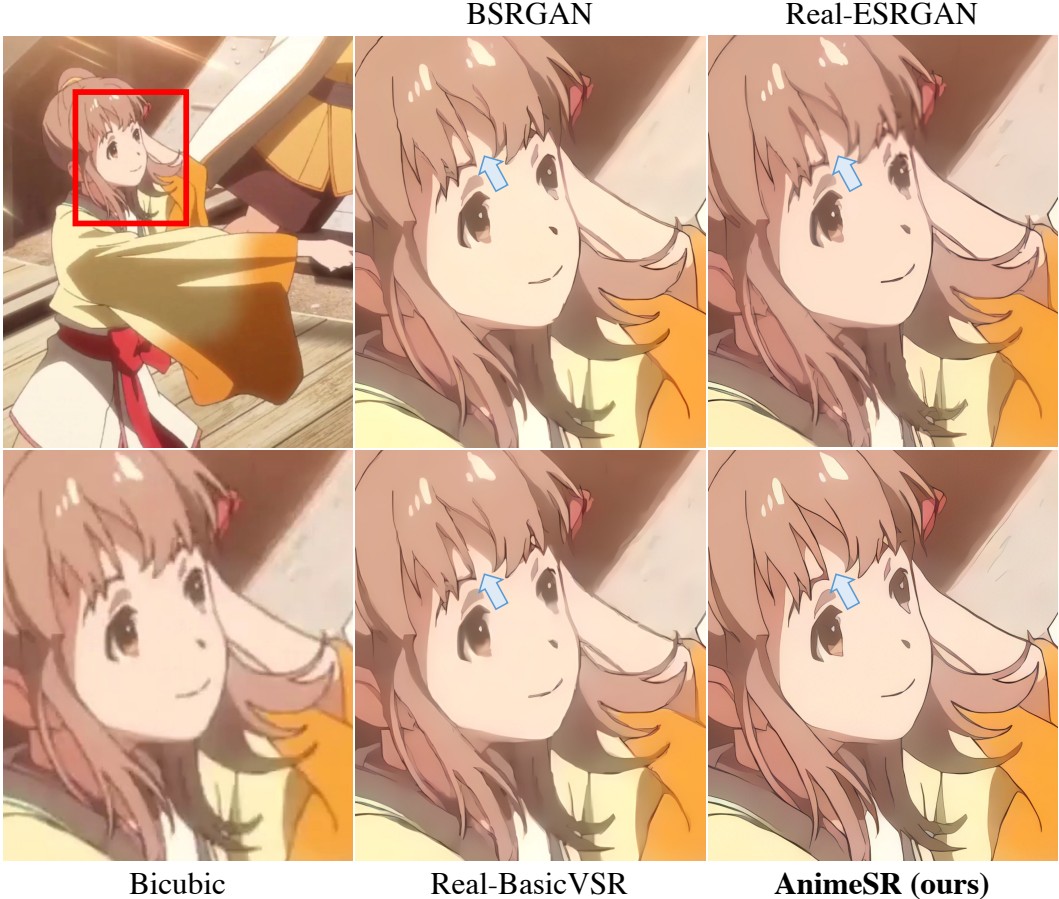

BSRGAN       Real-ESRGAN

Bicubic       Real-BasicVSR       **AnimeSR (ours)**

Figure 5: Qualitative comparison. Our method can recover cleaner lines with fewer artifacts (*e.g.*, around hair and eyes). **Zoom in for best view**

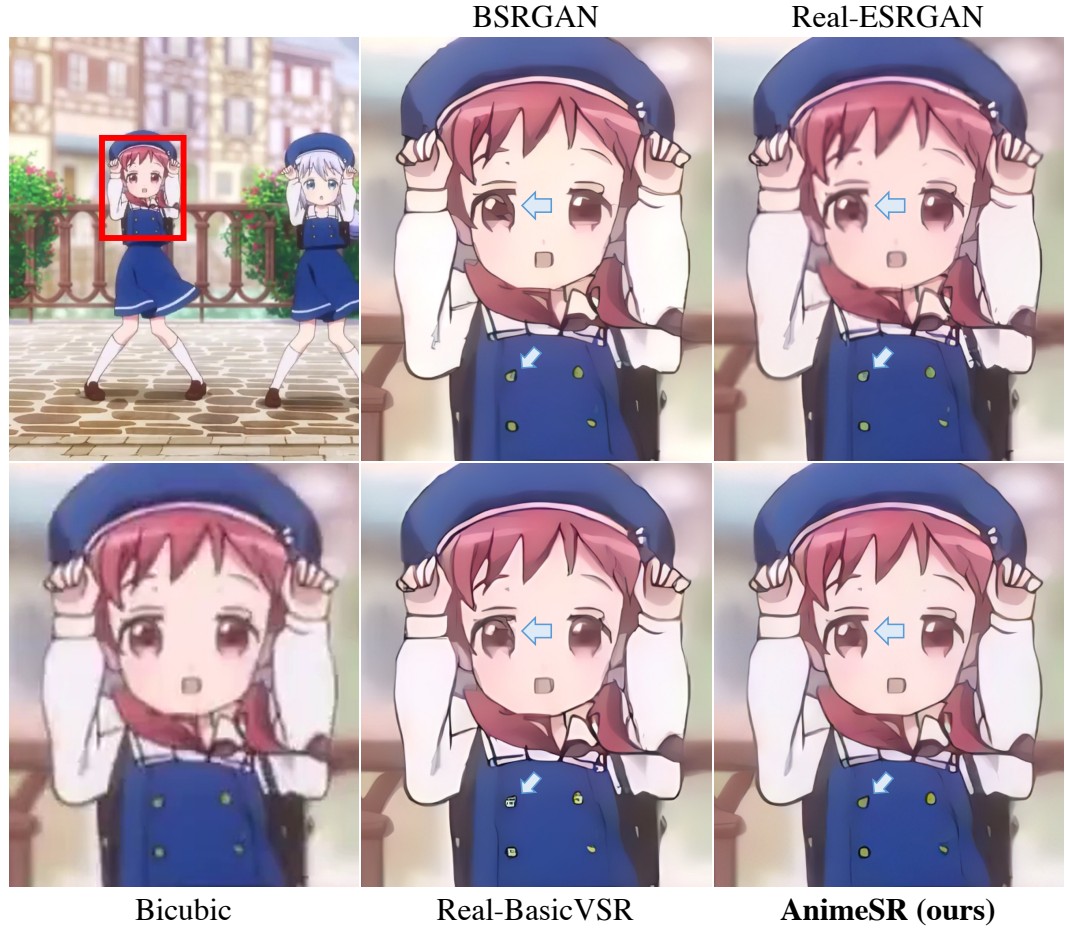

Figure 6: Qualitative comparison. Our method can recover cleaner lines with fewer artifacts (*e.g.*, around buttons on the cloth and eyes). **Zoom in for best view**

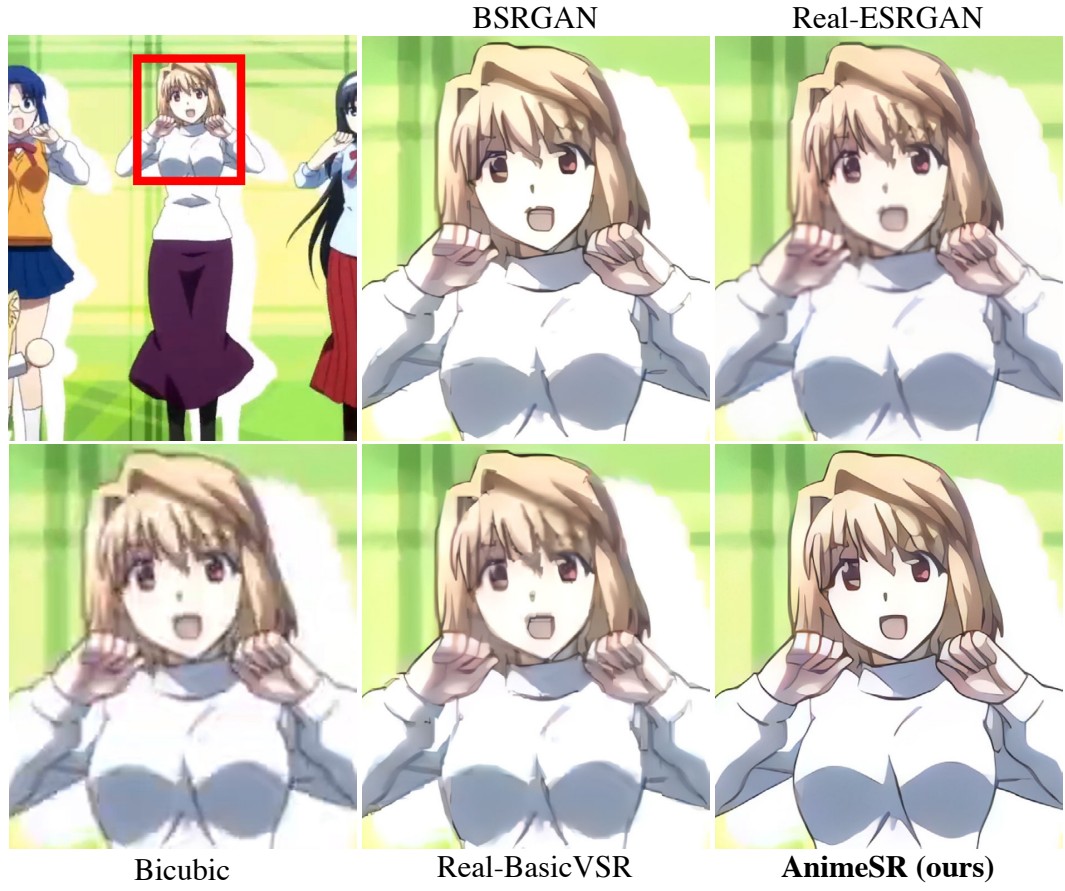

Figure 7: Qualitative comparison. Our method can recover cleaner lines with fewer artifacts (*e.g.*, around hairs and clothes). **Zoom in for best view**

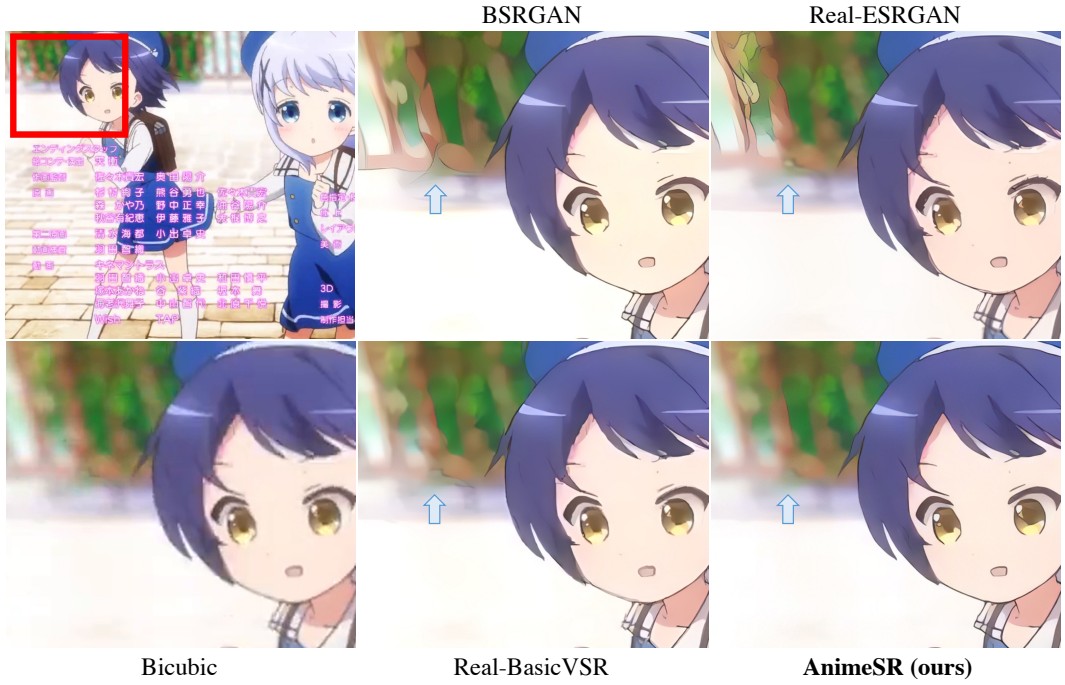

Figure 8: Qualitative comparison. Our method can produce more natural outputs in the eye with fewer artifacts and noises. For the background railing, our method is able to retain the overall naturalness and coherency. **Zoom in for best view**

BSRGAN Real-ESRGAN

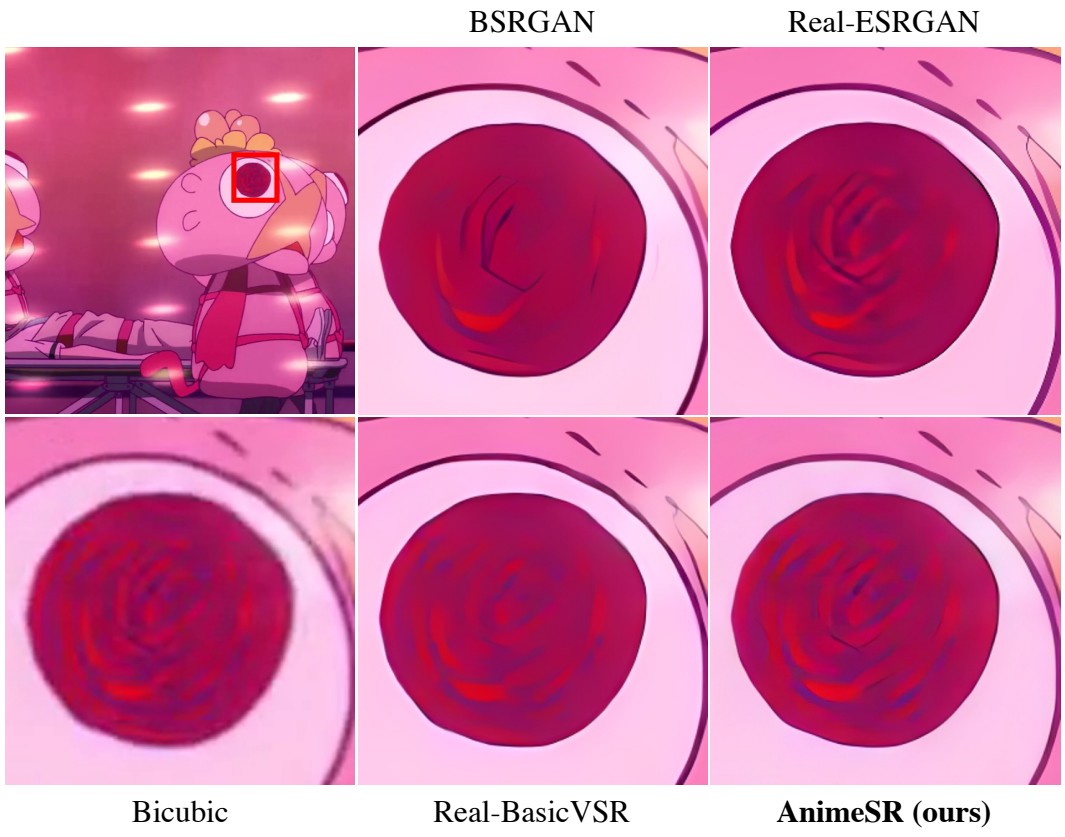

Bicubic Real-BasicVSR **AnimeSR (ours)**

Figure 9: Qualitative comparison. Our method can restore more texture details. **Zoom in for best view**

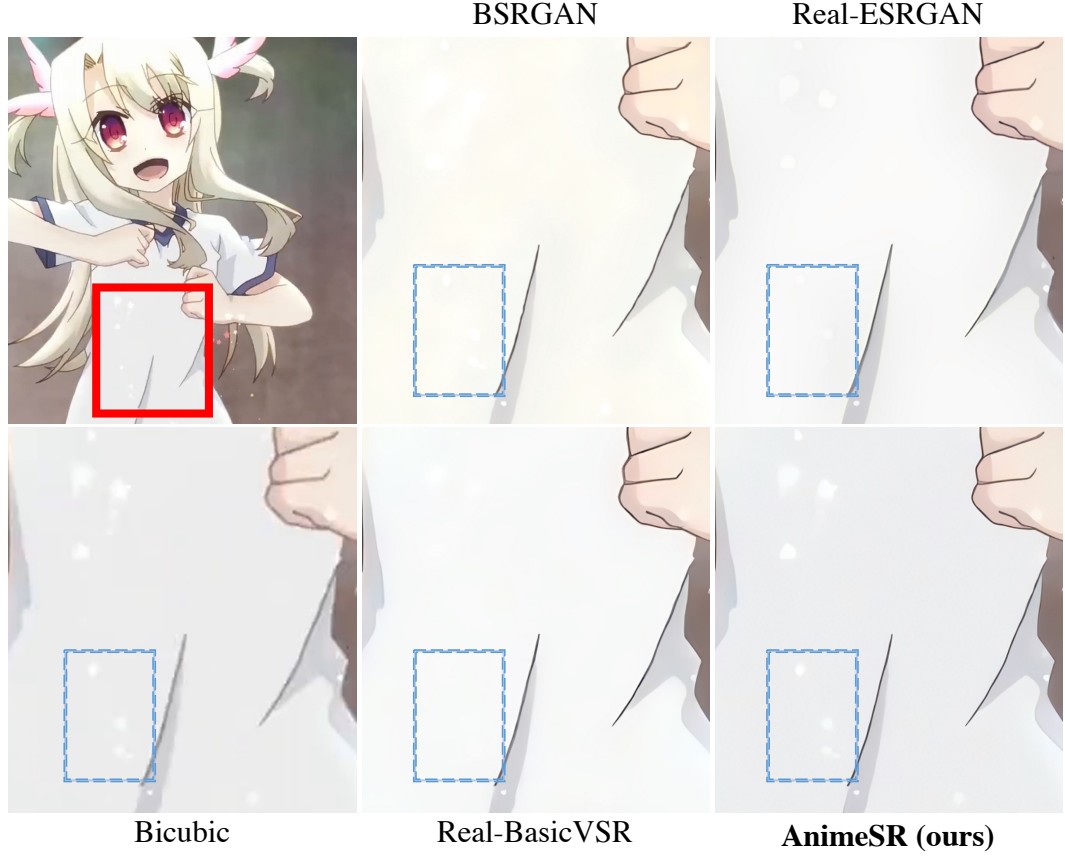

BSRGAN          Real-ESRGAN

Bicubic          Real-BasicVSR          **AnimeSR (ours)**

Figure 10: Qualitative comparison. Our method can restore and retain the bright spots while other methods impair it as noise. **Zoom in for best view**

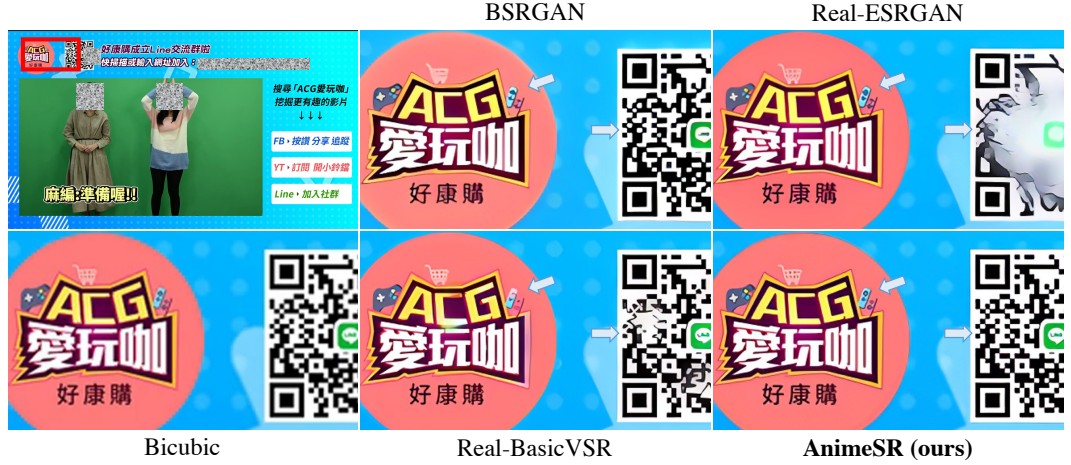

BSRGAN          Real-ESRGAN

Bicubic          Real-BasicVSR          **AnimeSR (ours)**

Figure 11: Qualitative comparison. Our method can restore more texture details for the QR code and the Joy-Con icon. **Zoom in for best view**

BSRGAN          Real-ESRGAN

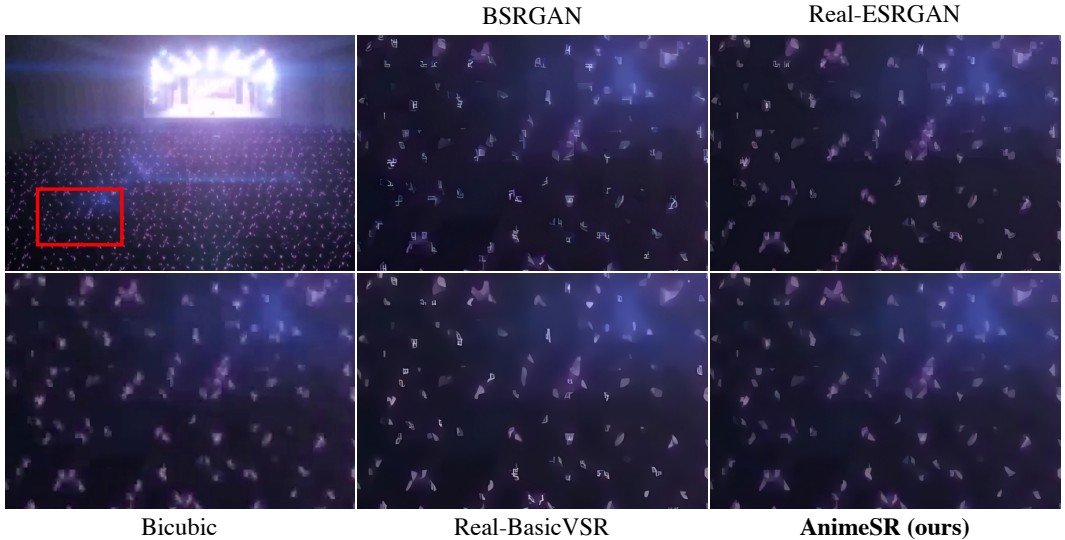

Bicubic          Real-BasicVSR          **AnimeSR (ours)**

Figure 12: Qualitative comparison. Our method can retain the overall naturalness. While other methods over-sharpen the crowd and produce artifacts. **Zoom in for best view**

BSRGAN                  Real-ESRGAN

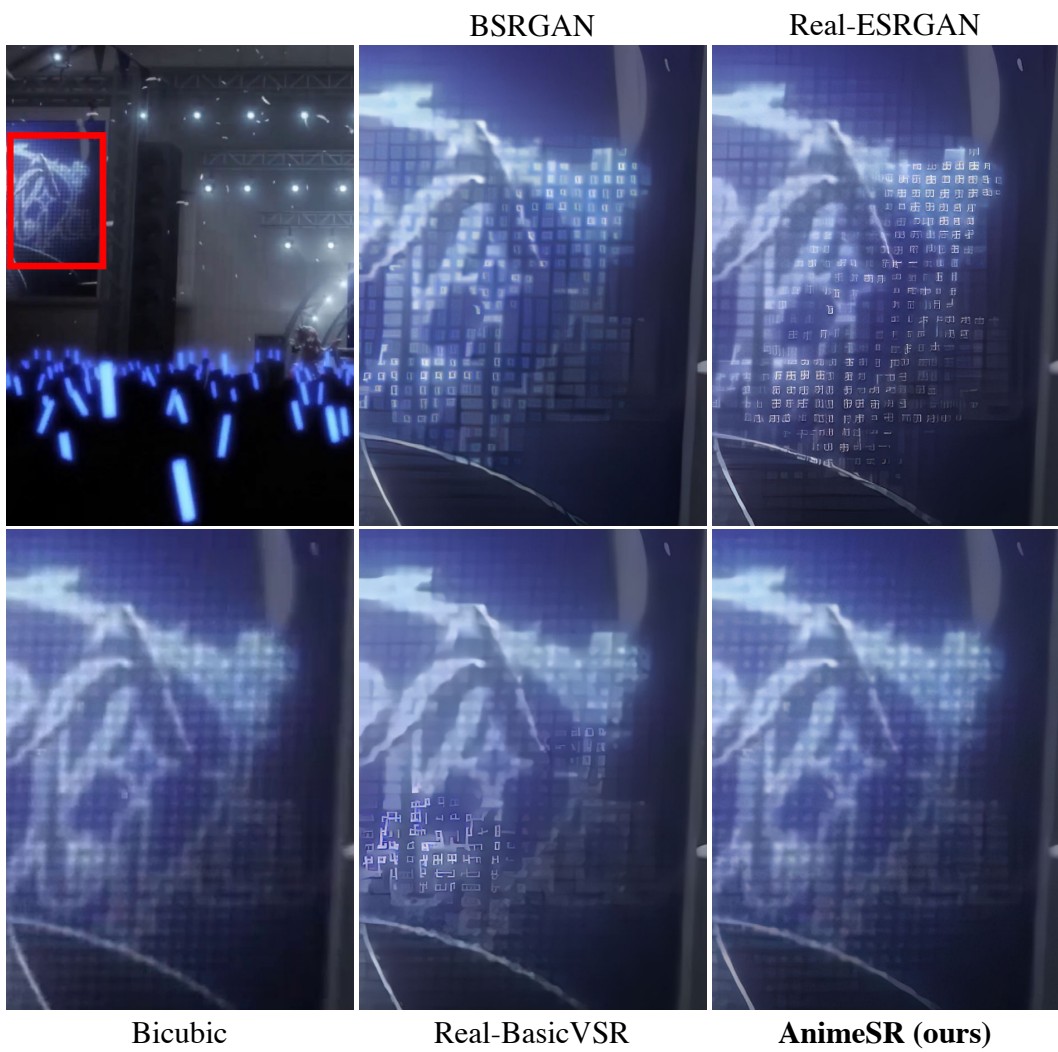

Bicubic            Real-BasicVSR          **AnimeSR (ours)**

Figure 13: Qualitative comparison. Our method can retain the overall naturalness. While other methods over-sharpen the grid texture and produce artifacts. **Zoom in for best view**

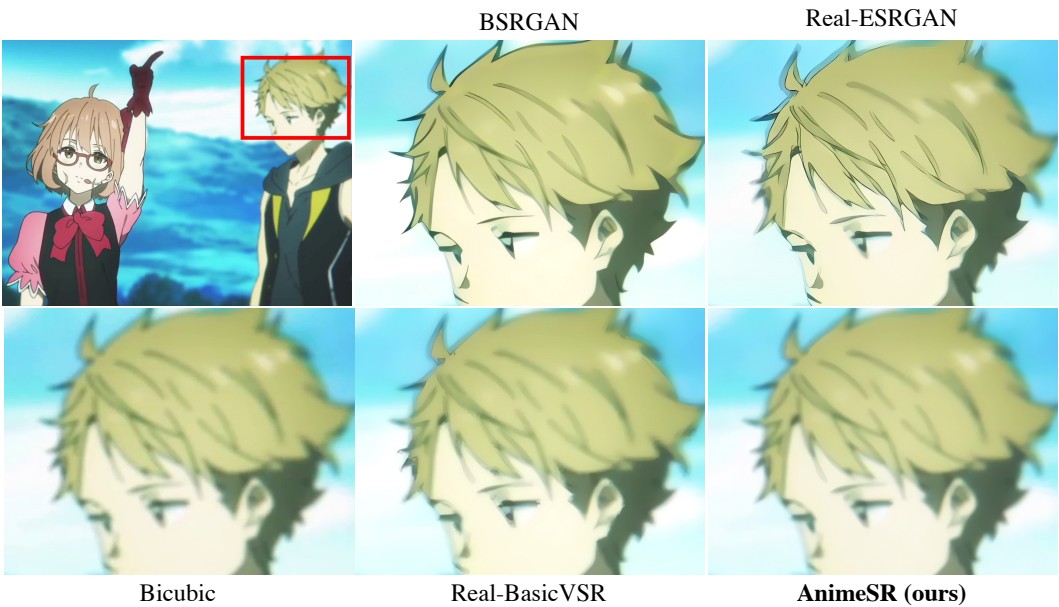

BSRGAN Real-ESRGAN

Bicubic Real-BasicVSR **AnimeSR (ours)**

Figure 14: Qualitative comparison. For the artificially blurred area in the image (*i.e.*, the boy), our method keeps the original style and does not over-sharpen it. While other methods over-sharpen it, destroying the effect of shallow depth of field. **Zoom in for best view**