# OpenReview forum: "AnimeSR: Learning Real-World Super-Resolution Models for Animation Videos"
_NeurIPS.cc/2022/Conference — NeurIPS 2022 Accept_

### Official Review · Reviewer_UxQQ · 2022-07-05

**Rating:** 7
**Confidence:** 3
**Soundness:** 3 good
**Presentation:** 2 fair
**Contribution:** 3 good

**Summary:**

This paper aims to develop an effective and efficient real-world video super-resolution (VSR) method to restore real-world low-quality animation videos. The authors first proposed to learn basic operators with tiny neural networks (with 2 or 3 conv layers) to incorporate the learned networks into the degradation synthesis process. Secondly, the authors propose a new dataset AVC which is a large-scale high-quality animation video dataset for training and evaluation. The authors perform experiments on AVC dataset and demonstrate superior performance compared to state-of-the-art models.

**Questions:**

1. Does the proposed input-rescaling strategy only work for animation videos? Since the authors anticipate that the input-rescaling strategy will not work well for natural videos due to textures, ablation studies on a texture could be helpful to quantitively investigate the scalability of this work.

2. How to determine the best rescaling factor of inputs for each video? Since the optimal rescaling factor varies on each video, pseudocode to determine the rescaling factor can help understand the overall process.


**Limitations:**

Societal Impact: There is no significant negative societal impact of this research.

**Strengths And Weaknesses:**

Strength: Instead of using one large neural network for the whole degradation process or classic basic operators without any learning capability, this paper proposes a novel method to learn basic operators with tiny neural networks. Meaningful discussion is also a strength of this paper (e.g. discussion on input-rescaling strategy, visualization of learned LBOs). Further, this paper presents a new dataset that contains more than a total of 50,000 frames, which is one of the main contributions of this paper.

Weakness: The writing can be improved. Section 4.2 is particularly confusing. The novelty of the approach seems limited. In particular, many components presented in this work are simple modifications from existing work (e.g. combining the classic basic operators and existing neural-network-based operators). Further, the proposed supervised manner of training heavily relies on an input-rescaling strategy, which restricts the scalability of work to the animation videos.

---

> ### Author Response · Authors · 2022-08-02
> **Author Response for the Reviewer UxQQ**
>
> We thank Reviewer UxQQ for the valuable reviews.
>
> ---
>
> **`Q1:` Limited novelty. Many components are simple modifications from existing work (e.g., combining the classic operators and neural-network-based operators).**
>
> 1. We introduce a new degradation synthesis paradigm. Although methods like BSRGAN and Real-ESRGAN employ classic basic operators to form a complex degradation model. The degradation model still could not cover the real-world degradation space. With the learnable basic operator, the degradation space can be largely expanded and can cover more real degradations. In addition, the learnable basic operators are quite different from existing neural-network-based operators. It is challenging to adopt one large neural network to learn the whole degradation process and the entire complicated degradation distribution. In this paper, we use LR - pseudo HR pairs to train the learnable basic operators by taking into account the properties of animation videos. Our experiments also show excellent results in simulating real-world degradations and restoring practical low-quality videos. Reviewer 7uTp and Reviewer Pf9p also acknowledge its novelty and promising values.
>
> 2. Besides the degradation model, we also collect a high-quality animation video dataset, and propose a compact VSR network based on the characteristics of animation videos. The network helps to restore real-world LQ animation videos effectively and efficiently ($13.7$× faster than BSRGAN and $5.9$× faster than RealBasicVSR). The high efficiency of our proposed compact VSR network makes our AnimeSR a more practical animation VSR model. Furthermore, our paper is also the first comprehensive work about animation VSR.
>
> ---
>
> **`Q2:` The training heavily relies on the input-rescaling strategy, which restricts the scalability of work to the animation videos.**
>
> In this work, we focus on the animation video SR task. We investigate the underlying characteristics of animation videos and observe that rescaling inputs of animation videos achieve a better balance of detail enhancement and artifact elimination. Based on this observation, we propose the input-rescaling strategy for animation videos. Due to the different characteristics of natural videos and animation videos, we agree with the reviewer that the input-rescaling strategy cannot be directly applied to natural videos.
>
> However, the strategy of combining learnable basic operators and classic basic operators is not restricted to animation videos. Learnable basic operators can expand the degradation space and can cover more real degradations. The input-rescaling strategy proposed in this paper is just one of the strategies specially designed for animation videos. We will discuss its effectiveness on textures in natural scenes in the revised version. Other strategies are left to investigate for natural videos.
>
> ---
>
> **`Q3:` How to determine the best rescaling factor for each video?**
>
> The input-rescaling strategy is **only** used in learning the learnable neural operators. The best rescaling factor is selected with a combination of algorithms and manual selection/verification. However, the algorithms' performance is not always satisfactory, especially for distinguishing the fine-grained artifacts. Thus, manual selection/verification is necessary.
>
> The details of the rescaling factor selection are as follows.
>
> 1. Select patches with textures and edges, because unpleasant artifacts are most likely to appear in those areas. The selection is based on edge detection.
> 2. Evenly sample rescaling factors from (0, 1] with an interval of 0.1. For each rescaling factor, LR patches are rescaled and are then sent to the BasicOP-Only VSR model to get the SR results.
> 3. Sort the rescaling factor based on the number of artifacts.
> Selecting results with the fewest artifacts is very challenging, we have tried several image assessment algorithms (e.g., NIQE, hyperIQA), but do not get a satisfying result. Empirically, we compare simple image statistics for the assistant, followed by a manual selection. Specifically, we calculate image gradients of LR and SR patches, and then sort the rescaling factors according to the similarity between the LR's normalized gradients and SR's. The intuition is that the unpleasant artifacts in animation videos, such as "hollow lines" artifacts and unwanted textures (Figure 5 in the main paper) usually produce a difference in image gradients.
> 4. Based on the sorting from Step 3, we manually select the best rescaling factor and pseudo HR according to human perception.
>
> Note that models trained with a few learnable neural operators ($3$ LQ videos in this work) can generalize well to a lot of real-world videos. So, even with some manual selections, the method is still valuable and does not take much time.
>
> ---
>
> **`Q4:` The writing can be improved. Sec. 4.2 is confusing.**
>
> Thanks for your suggestion. We compressed the contents in Section 4.2 in the submission version. We will improve it.

---

### Official Review · Reviewer_Pf9p · 2022-07-11

**Rating:** 6
**Confidence:** 4
**Soundness:** 2 fair
**Presentation:** 3 good
**Contribution:** 3 good

**Summary:**

This paper aims to improve blind video super-resolution (VSR) for animation videos. It combines two categories of degradation synthesis processes (classic basic operator and neural networks) to overcome both limitations. Besides, the authors build a new high-quality animation video dataset for animation VSR. Moreover, the paper designs an efficient network structure for VSR based on sliding window structures and multi-scale design. The experiments reveal that the proposed model outperforms other state-of-the-art models and validate the effectiveness of the model design.

**Questions:**

1. The author should present how to use the “input-rescaling strategy” in detail and discuss the time-consuming issue.
2. The author should differentiate the proposed method from FRVSR, especially the sliding window design.

**Limitations:**

The authors adequately addressed the limitations and potential negative societal impact of their work.

**Strengths And Weaknesses:**

Strengths:
1. It is interesting and seems promising to combine classic basic operators and neural networks together and learn basic operators by neural networks.
2. The authors build a large-scale high-quality animation video dataset for animation VSR.
3. There are plenty of experiments in the paper to validate the effectiveness of the model design. The experiments demonstrate that the proposed methods perform better than other state-of-the-art models and the combination of classic basic operators and neural networks is beneficial.

Weaknesses:
1. The use of the “input-rescaling strategy” is not demonstrated well. How do the users manually select the pseudo HR? It seems time-consuming to do it.
2. The intuition behind sliding window structure and multi-scale design is not demonstrated well. How to distinguish it from FRVSR? FRVSR is also based on sliding windows.

---

> ### Author Response · Authors · 2022-08-02
> **Author Response for the Reviewer Pf9p**
>
> We thank Reviewer Pf9p for the valuable reviews. Thanks for the acknowledgment on the interesting and promising approach of combining classic basic operators and tiny neural networks together. Our responses are as follows.
>
> ---
>
> **`Q1:` The use of the “input-rescaling strategy” is not demonstrated well. How do the users manually select the pseudo HR? It seems time-consuming.**
>
> Sorry for the unclear description. We provide more details about the use of input-rescaling strategy and the pseudo HR selection.
>
> **1. The input-rescaling strategy is used to learn the learnable neural operators.**
>
> The overall pipeline of training AnimeSR can be briefly summarized as follows:
>
> **Step 1: Construct the degradation model by combining classic basic operators and learnable neural operators.**
> The learnable operators are learned from a few real-world low-quality videos (**$3$ low-quality videos** in this work and those videos are not overlapped with AVC-RealLQ). With the input-rescaling strategy, we can generate pseudo HR frames, and then adopt the low-resolution and pseudo HR pairs to train the learnable operators.
>
> **Step 2: Train the SR network with the AVC-Train dataset.**
> On the high-quality AVC-Train dataset, we synthesize LR frames with the degradation model obtained from Step 1. We can then train the SR network. In Step 2, we do not use the input-rescaling strategy.
>
> **2. Select the best rescaling factor and pseudo HR.**
> We conduct the rescaling factor selection with a combination of algorithms and manual selection/verification. However, the algorithms' performance is not always satisfactory, especially for distinguishing the fine-grained artifacts. Thus, manual selection/verification is necessary.
>
> The details of the rescaling factor selection are as follows. The SR results corresponding to the best rescaling factor are regarded as pseudo HR.
>
> 1. Select patches with textures and edges, because unpleasant artifacts are most likely to appear in those areas. The selection is based on edge detection.
> 2. Evenly sample rescaling factors from (0, 1] with an interval of 0.1. For each rescaling factor, LR patches are rescaled and are then sent to the BasicOP-Only VSR model to get the SR results.
> 3. Sort the rescaling factor based on the number of artifacts.
> Selecting results with the fewest artifacts is very challenging, we have tried several image assessment algorithms (such as NIQE, hyperIQA), but do not get a satisfying result. Empirically, we compare simple image statistics for the assistant, followed by a manual selection. Specifically, we calculate image gradients of LR patches and SR patches, and then sort the rescaling factors according to the similarity between the LR's normalized gradients and SR's. The intuition is that the unpleasant artifacts in animation videos, such as "hollow lines" artifacts and unwanted textures (Figure 5 in the main paper) usually produce a difference in image gradients.
> 4. Based on the sorting from Step 3, we manually select the best rescaling factor and pseudo HR according to human perception.
>
> **3. The selection is only performed for a few videos.**
> Models trained with a few learnable neural operators can generalize well to a lot of real-world videos. In this work, the learnable neural operators from $3$ LQ videos can well generalize to a large number of real-world LQ videos. So, even with some manual selections, the method is still valuable and does not take much time. Note that the manual selection is only performed in learnable neural operators. The training dataset, whose size is much larger, does not involve the selection.
>
> ---
>
> **`Q2:` The intuition behind sliding window structure and multi-scale design is not demonstrated well. How to distinguish it from FRVSR? FRVSR is also based on sliding windows.**
>
> 1. Unidirectional recurrent architecture can only propagate the information from past frames and cannot propagate the information from future frames. Bidirectional recurrent is a feasible solution to get future frame information. But it takes double time and makes it unpractical. We combine unidirectional recurrent architecture and sliding window. The sliding window design only needs to look forward one more frame to obtain future information, while can still have the efficiency of unidirectional recurrent architecture.
> 2. As analyzed in Sec. 4.1 in the main paper, rescaling inputs of animation videos achieves a better balance of detail enhancement and artifacts elimination. Motived by this, we also adopt multi-scale architecture in animation VSR, which can leverage different rescaling inputs implicitly inside the network. The network can learn to utilize the features with the best scale levels.
> 3. We would like to kindly point out that FRVSR is not based on sliding windows. Instead, FRVSR is a unidirectional recurrent architecture and does not utilize future frames. In contrast, our architecture further incorporates the sliding window and multi-scale designs.

---

> > ### Comment · Reviewer_Pf9p · 2022-08-08
> > **Concern about Q2**
> >
> > Thank the authors for the response.  It addresses most of my concerns, especially Q1. I still have some questions about Q2.
> > 1. I think FRVSR is also based on sliding windows and the sliding windows only include the previous lr frame, the previous hr frame and the current lr frame. The architecture is unidirectional recurrent.
> > 2. In the Figure 6a, may i know whether $LR_{t-1}$, $LR_{t}$, $LR_{t+1}$ and $SR_{t-1}$ are concatenated together as the input of the multi-scale recurrent block?

---

> > > ### Author Response · Authors · 2022-08-09
> > > **Further Response to Q2**
> > >
> > > Thanks for your feedback :-)
> > >
> > > **1.** We think that FRVSR is not a strict sliding-window-based method. Sliding-window-based methods (such as EDVR[1], TOFlow[2], TDAN[3], VESPCN[4]) typically take several LR frames as inputs for restoration, usually including past frames, the current frame, and the future frames. FRVSR indeed uses the $LR_{t-1}$ frame, but for the flow estimation purpose. FRVSR does not take $LR_{t-1}$ as the input for restoration directly. Besides, it does not use future frames.
> > >
> > > That is the reason why we do not think FRVSR is a strict sliding-window-based method.
> > > But from a broader view, we *agree with the reviewer* that FRVSR can be regarded as a sliding-window-based method, since FRVSR uses both $LR_{t-1}$ and $LR_t$ for each time step $t$.
> > >
> > > We will clarify the above discussions.
> > >
> > > We then want to highlight that the emphasis of our method is different from FRVSR. FRVSR is the unidirectional recurrent structure and only uses the past and the current frames, while our method also wants to utilize the future frames to boost the performance.
> > > Leveraging the future frame information is the main feature of bidirectional recurrent structure and sliding windows (with future frames).
> > > Our method takes advantage of both the efficiency of the unidirectional recurrent structure and the effectiveness of future frames from sliding windows (with future frames).
> > >
> > > Besides, our method is also different from FRVSR in the following aspects.
> > >
> > > 1). We further adopt the multi-scale design. Due to the characteristics of animation videos, we observe that rescaling inputs of animation videos can achieve a better balance of detail enhancement and artifact elimination. Thus, we incorporate the multi-scale design and leverage different rescaling inputs implicitly inside the network.
> > >
> > > 2). We do not use flow estimation and warp components in our method. This is mainly due to the inference speed consideration. From our experiments, we do not observe an apparent performance drop when we remove flow estimation and warp.
> > >
> > > 3). Our experimental results show that directly applying FRVSR network structure is inferior to our structure.
> > >
> > > |Network Structure |    MANIQA score **↑ (high is better)**|
> > > | ----------- | ----------- |
> > > |FRVSR|                0.3583|
> > > |Ours              |   **0.3839**|
> > >
> > > **2.** Yes, $LR_{t-1}$, $LR_t$, $LR_{t+1}$ and $SR_{t-1}$ are concatenated together as the input of the multi-scale recurrent block. $SR_{t-1}$  is first downsampled by pixel-unshuffle operation to match the spatial resolution with other frames.
> > > We use concatenation as it can achieve fast inference speed while utilizing extra information and obtaining good performance.
> > >
> > > Hope those clarifications can address your concerns. Please let us know if you still have any unclear parts of our work. Thanks :-)
> > >
> > > Best,
> > >
> > > Paper 1447 Authors
> > >
> > > [1] EDVR: Video Restoration with Enhanced Deformable Convolutional Networks
> > >
> > > [2] TOFlow: Video Enhancement with Task-Oriented Flow
> > >
> > > [3] TDAN: Temporally-Deformable Alignment Network for Video Super-Resolution
> > >
> > > [4] VESPCN: Real-Time Video Super-Resolution with Spatio-Temporal Networks and Motion Compensation

---

> ### Author Response · Authors · 2022-08-08
> **Further discussions with Reviewer Pf9p**
>
> Dear Reviewer Pf9p,
>
> We thank you for the precious review time and valuable comments. We have provided corresponding responses, which we believe have covered your concerns. We hope to further discuss with you whether or not your concerns have been addressed. Please let us know if you still have any unclear parts of our work. Thanks :-)
>
> Best,
>
> Paper 1447 Authors

---

### Official Review · Reviewer_7uTp · 2022-07-12

**Rating:** 5
**Confidence:** 4
**Soundness:** 3 good
**Presentation:** 3 good
**Contribution:** 3 good

**Summary:**

This paper proposes an animation video super-resolution method.
First, the authors build the proposed AVC dataset by collecting low-quality videos and by generating pseudo-HR videos. Pseudo-HR videos are obtained by rescaling low-quality videos and super-resolving them with a conventional SR method.
Second, the degradation of animations are modeled by the combination of conventional corruption artifacts and learnable convolutional layers. The authors train their degradation model with the AVC dataset LR & pseudo-HR pairs.
Third, the proposed model architecture, window-based RNN is trained by the pseudo-HR dataset and the generated LR videos.


**Questions:**

L172-174 and Fig. 5.
I don’t exactly understand what the process of input-rescaling strategy is. Is input low-quality video frame downsampled and then super-resolved from a BasicOP-Only method?

Figure 9

Is this a result of equation (2)-like cascade of blur, noise, and LBO?
In equation (2), does two LBO share weights?

L318

Blur and noise are explicitly modeled in equation (2) even without LBO. Why does LBO learn to blur and noise images?

L306

Is one LBO learned per a video?

**Limitations:**

Limitations discussed in the manuscript.

**Strengths And Weaknesses:**

[Strengths]

This paper tackles the animation super-resolution problem by observing the distribution of animation and natural videos differ. The observation is validated by comparing the SoTA methods and the proposed method on animations.
In order to handle the unknown degradation artifacts that are hard to model, the proposed method combines the known operations (blur, noise, FFMPEG) and learnable operations (conv layers). This is a novel and effective approach.

Also, the authors construct pseudo-HR data from their observations which exhibits validity through experimental results.

[Weaknesses]

However, I wonder what the main principle is in the collected & generated dataset.
The video selection seems to be manual (which is fine) and the rescaling factor selection seems to rely on human perception. Were there any analyses that are not shown in the paper?
The authors use NIQE and MANIQA metrics. Any correlations with those metrics?
If the selection process is manual, it doesn’t seem to be a reproducible and scientific approach.

---

> ### Author Response · Authors · 2022-08-02
> **Author Response for the Reviewer 7uTp (Part 3/3)**
>
> **`Q6:` Is Figure 9 a result of equation (2)-like cascade of blur, noise, and LBO? In equation (2), does two LBO share weights?**
>
> 1. Fig. 9 is not a result of equation (2). Fig. 9 visualizes what the $3$ LBOs have learned and how they differ from classic basic operators like classical Gaussian blur and Gaussian noise. All the LQ patches in Fig. 9 are synthesized from the same HQ patch by using **only** one LBO or **only** the Gaussian blur/noise, instead of a cascade result.
> We train $3$ different LBOs and their corresponding synthesized LQs are visualized.
>
> 2. The two LBO in Equation (2) are random selections from an LBO pool ($3$ in this paper). So they are can be different (with different weights), and it is also possible to select the same LBO in Equation (2).
>
> ---
>
> **`Q7:` Blur and noise are explicitly modeled in equation (2) even without LBO. Why does LBO learn to blur and noise images according to L318?**
>
> In L315-L318, the paper describes: "As shown in Fig. 9, the degradations learned by LBO are **pretty different from classic basic operators, e.g., blur and noise**. It seems that LBO captures a mixture of blur and noise. We can also observe **color jitter around lines** in the LBO-1 neural operator, which is common in highly-compressed LQ videos."
>
> We do not mean that LBO learns to blur and noise images. Instead, the degradations learned by LBO are **pretty different** from classic basic operators. Visually, it seems to be a mixture of blur and noise. Here, the blur and noise are not actually Gaussian blur and Gaussian noise, but a vague human perception. This is not an accurate description and we will improve it.
>
> ---
>
> **`Q8:` Is one LBO learned per video?**
>
> Yes, one LBO is learned per video. We learn $3$ LBO in this paper, and it shows great generalizability to a large number of real-world LQ videos (*e.g.*, videos in AVC-RealLQ).

---

> > ### Comment · Reviewer_7uTp · 2022-08-09
> > **Post-rebuttal response**
> >
> > I thank the authors for the detailed comments and explanations.
> >
> > Most of the information I wanted to know is clarified in the rebuttal.
> > However, I believe it is worth adding such detailed descriptions in the manuscript to avoid any potential confusion for readers.
> >
> > While I lean towards a positive impression, one remaining concern is that I still think this method requires human labor in rescaling factor selection.
> > Although that makes the proposed method work well, it is a limiting factor making it hard to extend to other datasets or settings.

---

> > > ### Author Response · Authors · 2022-08-09
> > > **Thanks for your response**
> > >
> > > Thanks for your feedback.
> > >
> > > We are glad that the detailed comments and explanations can clarify the unclear parts. We will add those detailed descriptions to the manuscript.
> > >
> > > We agree with the reviewer that the human labor in the rescaling factor selection is a limitation. As explained above, such manual selections are only conducted on three LQ videos and then the learned neural operators can well generalize to a large number of real-world LQ videos. So, the method is still valuable and does not take much time.
> > >
> > > We make the first attempt to explore the underlying characteristics among different rescaling factors and involve manual selection in our method. We believe there will be better designs for rescaling factor selection, which can be left a future investigation.

---

> ### Author Response · Authors · 2022-08-02
> **Author Response for the Reviewer 7uTp (Part 2/3)**
>
> **`Q3:` The rescaling factor selection seems to rely on human perception.**
>
> We conduct the rescaling factor selection with a combination of algorithms and manual selection/verification. However, the algorithms' performance is not always satisfactory, especially for distinguishing the fine-grained artifacts. Thus, manual selection/verification is necessary.
>
> The details of the rescaling factor selection are as follows.
>
> 1. Select patches with textures and edges, because unpleasant artifacts are most likely to appear in those areas. The selection is based on edge detection.
> 2. Evenly sample rescaling factors from (0, 1] with an interval of 0.1. For each rescaling factor, LR patches are rescaled and are then sent to the BasicOP-Only VSR model to get the SR results.
> 3. Sort the rescaling factor based on the number of artifacts.
> Selecting results with the fewest artifacts is very challenging, we have tried several image assessment algorithms (such as NIQE, hyperIQA), but do not get a satisfying result. Empirically, we compare simple image statistics for the assistant, followed by a manual selection. Specifically, we calculate image gradients of LR patches and SR patches, and then sort the rescaling factors according to the similarity between the LR's normalized gradients and SR's. The intuition is that the unpleasant artifacts in animation videos, such as "hollow lines" artifacts and unwanted textures (Figure 5 in the main paper) usually produce a difference in image gradients.
> 4. Based on the sorting from Step 3, we manually select the best rescaling factor according to human perception.
>
> It is worth noting that one of the main contributions of our paper is the combination of classic operators and learnable neural operators, to get a more accurate degradation model. We then investigate how to learn such learnable neural operators. Based on the characteristics of animation videos, we propose the rescaling factor selection, which is the first attempt to explore the underlying characteristics among different rescaling factors. We believe there will be better designs for future investigation.
>
> Besides, models trained with a few learnable neural operators can generalize well to a lot of real-world videos. In this work, the learnable neural operators from $3$ LQ videos can well generalize to a large number of real-world LQ videos. So, even with some manual selections, the method is still valuable and does not take much time.
>
> ---
>
> **`Q4:` The authors use NIQE and MANIQA metrics. Any correlations with those metrics?**
>
> Both NIQE and MANIQA are non-reference image assessment metrics.
> They are correlated on a coarse scale, but usually have different scores on a finer scale.
>
> In detail, NIQE is based on natural scene statistics and hand-crafted features, while MANIQA is a learning-based approach and employs a powerful ViT to extract features. MANIQA is the winner solution of "NTIRE 2022 Perceptual Image Quality Assessment Challenge Track 2: No-Reference (NR)" and outperforms previous SoTA methods on various datasets.
>
> In this work, we use both NIQE and NAMIQA for reference.
>
> ---
>
> **`Q5:` L172-174 and Fig. 5, what is the process of input-rescaling strategy. Is input low-quality video frame downsampled and then super-resolved from a BasicOP-Only method?**
>
> Yes, the input LQ inputs are downsampled with different rescaling factors and then super-resolved from a BasicOP-Only method. BasicOP-Only method is trained with degradation models consisting of only classic operators (blur, noise and FFMPEG).
>
> Then, we adopt the selection process (as described in **Q3**) to select the best rescaling factor. The SR results under the best rescaling factor, namely the pseudo HR, are used to train the learnable operators.

---

> ### Author Response · Authors · 2022-08-02
> **Author Response for the Reviewer 7uTp (Part 1/3)**
>
> We thank the reviewer for the insightful reviews. Thanks for the acknowledgment on the novel and effective approach of combining the known operators (blur, noise, FFMPEG) and learnable operators (conv layers). Our responses are as follows.
>
> ---
>
> **`Q1:` Clarification on the dataset and overall pipeline.**
>
> We first clarify some misunderstandings about the dataset and the overall pipeline. Sorry for the unclear descriptions in the main paper. We will improve related descriptions.
>
> The **training partition** in the AVC dataset (AVC-Train) is constructed by collecting **high-quality** animation video clips.
> The AVC-RealLQ partition is only for **testing**, which includes real-world **low-quality** videos.
>
> The overall pipeline of training AnimeSR can be briefly summarized as follows:
>
> **Step 1: Construct the degradation model by combining known operators and learnable operators.**
> The learnable operators are learned from a few real-world low-quality videos (**$3$ videos** in this work and those videos are not overlapped with AVC-RealLQ). The input-rescaling strategy is used to generate pseudo high-resolution (HR) frames. We then adopt the low-resolution (LR) and pseudo HR pairs to train the learnable operators.
>
> *Note that*:
>
> **1)** We use the input-rescaling strategy and generate pseudo HR for only three videos, whose size is very small compared to the training dataset (AVC-Train) used in Step 2.
>
> **2)** Those learned operators from 3 videos can **well generalize** to a large number of real-world LQ videos (*e.g.*, videos in AVC-RealLQ).
>
> **Step 2: Train the SR network with the AVC-Train dataset.**
> The AVC-Train dataset only contains high-quality video clips. We synthesize low-resolution (LR) frames with the degradation model obtained from Step 1. With the synthesized LR and HR pairs, we can then train the SR network. As the degradation model from Step 1 can better capture real-world degradations, the SR network thus performs better in practical scenarios.
>
> *Note that*:
>
> **1)** In Step 2, we do not use the input-rescaling strategy. Thus there is no need to select the best rescaling factor.
>
> **2)** We adopt HR clips in AVC-Train and the synthesized LR clips for training. There is no pseudo HR in Step 2.
>
> ---
>
> **`Q2:` What the main principle is in the collected dataset?**
>
> The SR task relies on high-quality datasets (*e.g.*, DIV2K for image SR and REDS for video SR). Those datasets typically serve as the Ground-Truth, and we usually train SR networks by synthesizing low-resolution counterparts from those datasets.
>
> The existing datasets for animation-related tasks are of low quality and contain only single images or triplet frames. Thus, considering the requirements for animation video SR, we conclude the following principles for the collected AVC dataset.
>
> 1. **High quality**. We ensure the video quality by bit rate, frame resolution, and subjective quality. We also use an image assessment quality algorithm (hyperIQA) to ensure the quality of each frame.
> 2. **dynamic and meaningful scenes**. We select scenes with motions for video tasks. Static and simple scenes (*e.g.*, scenes with black or pure colors) are discarded.
> 3. **diversity and amount**. They are common principles and are considered for better performance and generalization.
>
> We first employ existing algorithms for **automatic filtering**, followed by **manual selection/verification**, as automatic filtering is not always perfect.
>
> The AVC-RealLQ dataset is constructed to fully evaluate the model’s ability in practical scenes and assess the generalizability of VSR methods.
>
> Note that the input-rescaling strategy and pseudo HR are only used for **learnable operators in the degradation model**. They are not related to the AVC dataset, and they are not used in training SR networks.

---

### Meta-Review · Area_Chair_oa38 · 2022-08-27

**Recommendation:** Accept
**Confidence:** Certain

**Metareview:**

The paper proposes a method for super-resolution of animation videos. The contribution is three-fold: a new approach to learned image degradations, a dataset of high-resolution animation videos, and a multiscale model architecture. The method demonstrated good empirical results while being substantially faster than prior approaches.

All reviewers are positive about the paper (although to a different degree) and mention that the proposed "learned basic operators" are interesting and new, the dataset is valuable, and the method is thoroughly evaluated and works well.

Overall, the paper is a solid application paper with some interesting new ideas and I recommend acceptance.  I highly encourage the authors to update the paper based on the discussions with the reviewers, in particular with the details on dataset creation and the rescaling factor.

**Award:**

No

---

### Decision · Program_Chairs · 2022-09-14

Accept